# High-resolution dataset of thermokarst lakes on the Qinghai-Tibetan Plateau

Xu Chen [a], Cuicui Mu [a, b, c, d*], Lin Jia [a], Zhilong Li [a], Chengyan Fan [a], Mei Mu [a], Xiaoqing Peng [a], Xiaodong Wu [b, e]

[a] Key Laboratory of Western China's Environmental Systems (Ministry of Education), College of Earth and Environmental Sciences, Lanzhou University, Lanzhou, 730000, China

[b] Cryosphere Research Station on Qinghai-Tibetan Plateau, State Key Laboratory of Cryospheric Science, Northwest Institute of Eco-Environment and Resource, Chinese Academy of Sciences, Lanzhou, 730000, China

[c] Southern Marine Science and Engineering Guangdong Laboratory (Zhuhai), 519000, China

[d] University Cooperation of Polar Research, Beijing, 100875, China

[e] University of Chinese Academy of Sciences, Beijing, 100049, China

*Corresponding to:* Cuicui Mu, mucc@lzu.edu.cn

**Abstract.**The Qinghai-Tibetan Plateau (QTP), the largest high-altitude and low-latitude permafrost zone in the world, has experienced rapid permafrost degradation in recent decades, and one of the most remarkable resulting characteristics is the formation of thermokarst lakes. Such lakes have attracted significant attention because of their ability to regulate carbon cycle, water, and energy fluxes. However,

the distribution of thermokarst lakes in this area remains largely unknown, hindering our understanding of the response of permafrost and its carbon feedback to climate change. Here, based on the Google Earth Engine platform, we examined the modern distribution (2018) of thermokarst lakes on the QTP using Sentinel-2A data; for the first time providing the true spatial distribution by using a resolution of 10 m with a relative error of 0–0.5. Results show that the total thermokarst lake area on the QTP is 1730.34

m km², accounting for approximately 4% of the total water area of lakes and ponds, and that overall thermokarst lake density is 12/100 m km². More specifically, the densities of thermokarst lakes in the land types of alpine desert steppe (16/100 km²) and barren land (17/100 km²) were larger than those of alpine meadows (13/100 km²), alpine steppe (11/100 km²), and wet meadow (11/100 km²). These findings provide a scientific foundation for future investigations into the effects of climate change on the

permafrost environment and carbon emissions from rapidly developing thermokarst landscapes. Data are made available as open access via the National Tibetan Plateau Data Center (Chen et al., 2021) with DOI:

10.11888/Geocry.tpdc.271205.

(https://data.tpdc.ac.cn/en/data/c0c05207-568d-41db-ab94-8610bdcdbbe5/)

Key words: Climate change; permafrost degradation; Qinghai-Tibetan Plateau; thermokarst lakes

**1. Introduction**

One of the most obvious characteristics of permafrost degradation is the formation of thermokarst terrains,

which is a special geomorphic process initiated by the degradation of ice-rich permafrost or the melting

of thick underground ice (Kozarski et al., 1998). In comparison with tectonic lakes, thermokarst lakes

are usually smaller, forming active lakes and ponds typically less than 0.5 m km² (Niu et al., 2014).

Despite their smaller areas, thermokarst lakes are important components of permafrost regions, as they

are known to greatly impact human infrastructure, hydrologic processes, and terrestrial and aquatic

biogeochemical cycles (Marsh et al., 2009; Kokelf and Jorgenson, 2013). The distribution of thermokarst

lakes can even affect the amount and chemical form of greenhouse gases released from permafrost

regions. Thus, thermokarst lakes should be taken into consideration in future climate change projections

(Vonk and Gustafsson, 2013).

Thermokarst lakes are abundant in Arctic permafrost regions, which play an important role in Arctic

ecosystems (Morgenstern et al., 2011; Muster et al., 2017). The total area of lakes and ponds in the

circum-Arctic permafrost region has been found to be $1.4 \times 10^6$ m km² (Muster et al., 2017), while in

four extensive latitudinal transects in Alaska, Eastern Canada, Western Siberia, and Eastern Siberia,

thermokarst lakes cover a total area of more than $2.3 \times 10^6$ m km² (~10% of the permafrost region in the

Northern Hemisphere). Moreover, to date it has been found that 643,304 thermokarst lakes are larger

than 0.01 m km², covering a total area of 118,182 m km² (Nitze et al., 2018). The lake distributions

differed significantly among Eastern Canada (13.4%), Western Siberia (6.1%), Eastern Siberia (1.6%),

and Alaska (2.9%) (Nitze et al., 2018). From 1999–2014, the net changes of lakes were -5.46% in western

Siberia, -0.62% in Alaska, -0.24% in eastern Canada, and +3.67% in east Siberian (Nitze et al., 2018).

In the future, changes in thermokarst lakes are expected to be highly diverse due to the spatial

heterogeneities in surface geology, geomorphology, permafrost extents, and ground ice conditions

(Riordan et al., 2006; Jones et al., 2011; Chen et al., 2014; Nitze et al., 2018).



As the largest middle-low latitude and high-altitude permafrost region, the Qinghai-Tibetan Plateau (QTP) occupies a vast area underlain by permafrost, which is estimated to be as high as $1.06 \times 10^6$ m km² (Zou et al., 2017). Owing to its middle-low latitude, the permafrost on the QTP is characterized as being relatively thin, but with thick active layers and high ground temperatures (Ran et al., 2018; Ran et al.,

2020). In recent decades, permafrost on the QTP has experienced obvious degradation, as is indicated by the increasing ground temperatures (Hu et al., 2019) and further thickening of active layers (Zhao et al., 2019). Meanwhile, accelerated formation of thermokarst terrains, including permafrost collapse, thaw slump, and thermokarst lakes, has also been observed (Luo et al., 2019; Huang et al., 2020; Mu et al., 2020b). Among these features, thermokarst lakes are of the highest concern. On the QTP, most

thermokarst lakes have been reported from the middle plateau (Niu et al., 2011), and in previous reports, much attention has been paid to thermokarst lakes for their ability to cause serious thermal erosion and permafrost thawing, leading to the instability of road embankments (Niu et al., 2011). Recently, it has also been recognized that these lakes can release considerable greenhouse gas into the air (Wu et al., 2014; Mu et al., 2016; Mu et al., 2020a). Climate warming is expected to lead to an increase in the number

of thermokarst lakes forming in continuous permafrost areas (Wang and Mi, 1993), yet the identification and investigation of thermokarst lakes has mainly been conducted only at the local scale (Niu et al., 2008). For example, in the Beiluhe River basin, located in the middle plateau with an area of 2513.6 m km², it was found that thermokarst lakes showed an increasing trend from 1969 to 2010 (Luo et al., 2015). Thus far, the distribution and changes of thermokarst lakes at the larger plateau scale remain unknown, and

there is an urgent need to establish a high-resolution dataset of thermokarst lakes on the plateau in order to provide better scientific data for Earth System Models.

In this study, we extracted water bodies using the normalized difference water index (NDWI) from a large number of Sentinel-2 data based on the Google Earth Engine (GEE) platform (Mcfeeters and S., 1996; Ouma and Tateishi, 2006; Xu, 2006). Because this method has been associated with the

overestimation of water bodies, we also used visual interpretation to calibrate the automatically extracted water vectors and imagery. In addition, field-based unmanned aerial vehicle (UAV) imagery and field data retrieved from the literature were used to verify the thermokarst lake distribution. We further examined the relationships between the distribution of thermokarst lakes and temperature, precipitation, active layer thickness, vegetation coverage, and normalized difference vegetation index (NDVI). The

main aims of this study were to: 1) establish a high-resolution dataset of thermokarst lake distribution on

the QTP, and 2) explore the effects of environmental factors on the distribution of thermokarst lakes.

## 2. Study area

The QTP is the largest and highest plateau in the world, with an area underlain by permafrost of

approximately $1.06 \times 106$ m km² (Zou et al., 2017). From 1980 to 2018, the air temperature, precipitation,

and soil water content in the permafrost region showed a significant increasing trend (Yang et al., 2019;

Zhao et al., 2019). The largest permafrost thickness was found to be ~128 m, and the storage of

underground ice is estimated to be $\sim 1.27 \times 10^4$ km3 of water equivalent (Cheng et al., 2019). The active

layer thickness on the QTP ranges from 100 to 400 cm, and the active layer thickness along the Qinghai-

Tibet Highway has increased by 19.5 cm/10a from 1982 to 2018 (Zhao et al., 2019). In relation to the

mean annual ground temperature (MAGT) in the permafrost area of the QTP, it has been found that -

3 ℃ < MAGT < -1.5 ℃ accounted for 30.4%, -1.5 ℃ < MAGT < -0.5 ℃ accounted for 22.1%, and -

0.5 ℃ < MAGT < 0.5 ℃ accounted for 22.6% of the permafrost regions (Zhao et al., 2019; Ran et al.,

2020). The total area of lakes (> 1 m km³) for the entire QTP is $5 \times 10^4$ m km², accounting for 1.67% of

the total land area (Zhang et al., 2019; Zhang et al., 2020). In recent decades, the area and number of

lakes on the QTP have increased extensively (Zhang et al., 2017); for example, the number and total area

of lakes greater than 1 m km² expanded from 1081 and $4 \times 10^4$ m km² in the 1970s to 1236 and 4.74 ×

$10^4$ m km² in the 2010s, respectively (Zhang et al., 2014), owing to increased precipitation, glacier

melting, permafrost degradation, and other changes in additional components of terrestrial water (Liu et

al., 2019).

## 3. Data resources

Sentinel-2A, launched by the European Space Agency (ESA) on June 23, 2015, carries the sensor known

as Multi-Spectral Instrument (MSI). MSI has 13 spectral bands covering the visible spectrum (VIS),

near-infrared (NIR), and short-wave infrared (SWIR) parts. Sentinel-2A has three spatial resolutions of

10, 20, and 60 m, and a revisit time of 10 days (Drusch et al., 2012; Li and Roy, 2017). The Sentinel-2

mission, organized by the Global Environment and Security Monitoring (GMES), uses a twin-satellite

system to capture multi-spectral high-resolution optical observations at high revisit frequencies for about

five days on the ground around the world. The system features the advantage of intensive continuous

monitoring of the Earth's surface. Since December 2015, data can be acquired through free download

from the ESA official website (https://scihub.copernicus.eu/).

Only data at 10-m resolution were used in this study, and Sentinel-2A imagery was from the   United

States Geological Survey (USGS) (https://earthexplorer.usgs.gov). Meteorological data including

temperature and precipitation were obtained from ERA5, which is the fifth generation the European

Centre for Medium-Range Weather Forecasts (ECMWF) atmospheric global climate reanalysis (ERA

Interim, http://apps.ecmwf.int/datasets/data/interim-full-daily). The meteorological data used in this

study have a spatial resolution of 0.5 ×0.5 °(Dee et al., 2011).

The permafrost data were derived from the permafrost distribution data set based on the improved land

surface temperature (LST) of the Moderate Resolution Imaging Spectroradiometer (MODIS) and the

simulated permafrost top temperature (TTOP) (Figure. 1) (Zou et al., 2017). The active layer thickness

data were derived from the dataset, which was established based on the melt-index and Stefan method

(Peng et al., 2018).

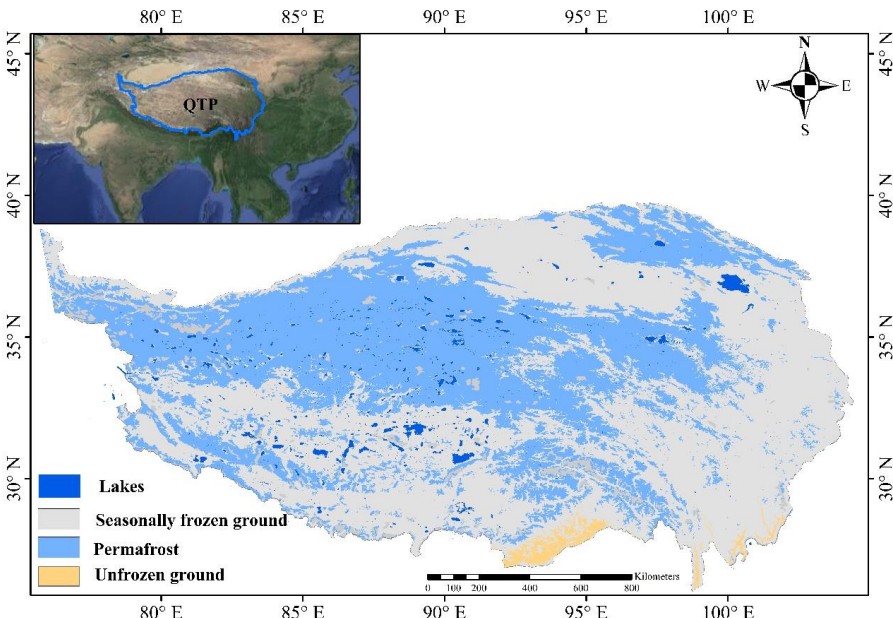

**Figure. 1 Distribution of permafrost on the Qinghai-Tibetan Plateau (QTP) (Zou et al., 2017)**


The time series of NDVI data were calculated using MODIS remote sensing images with a resolution of

m. The Digital Elevation Model (DEM) dataset with a resolution of 90 m was retrieved from the

Shuttle Radar Topography Mission (SRTM) terrain data, which were obtained from the International

Center for Tropical Agriculture (CIAT) using the interpolation algorithm (Reuter et al., 2007).

The MAGT of the QTP was retrieved from Ran et al. (2020), of which the MAGT data were established

using remote sensing data and the field-measured MAGT data of 233 boreholes (Ran et al., 2020). This

dataset has a spatial resolution of $1 \times 1$ km. Vegetation type data with a spatial resolution of $1 \times 1$ km in

permafrost areas of the QTP were obtained from Wang et al. (Wang et al., 2016), of which the land cover

types were classified into five types: swamp meadow, meadow, steppe, desert steppe, desert, and barren

land. Ground ice data were retrieved from a map of permafrost and ground ice in the Northern

Hemisphere (Brown, 2002) which describes the distribution of ground ice conditions. The subsurface ice

abundance of the topmost 20 m is divided into portions by the percentage of ice volume ($> 20\%$, 10–

20%, and $< 10\%$).


## 4. Methods

### 4.1 Research framework

The framework in this study comprises a collation of knowledge and formulation of the thermokarst lake

inventory specifications, as well as the data preprocessing completed using GEE, manual vectorization

of the thermokarst lakes, visual interpretation, and environmental factor extraction (Figure. 2).

*1. Specifications of the thermokarst lake inventory.* Literature relevant to the investigation and recording

of thermokarst lakes were collected. Various definitions and classifications of thermokarst lakes, as well

as the methods adopted previously for lake boundary extraction and assessment of the extent of lake

distributions, were summarized.

*2. Data preprocessing in Google Earth Engine (GEE).* Through the GEE platform (https://

earthEngine.google.com), NDWI values were used to automatically extract the overall total water body

of the QTP with a resolution of 10 m (Sentinel-2A) in 2018. At the same time, the extraction of

environmental factors was also carried out using GEE.

*3. Visual interpretation and manual vectorization of thermokarst lakes.* Inventory work was performed



in 2019, including lake boundary vectorization of the QTP. By comparing Sentinel-2 remote sensing

images from 2018, visual interpretation was used to correct the number and range of thermokarst lakes.

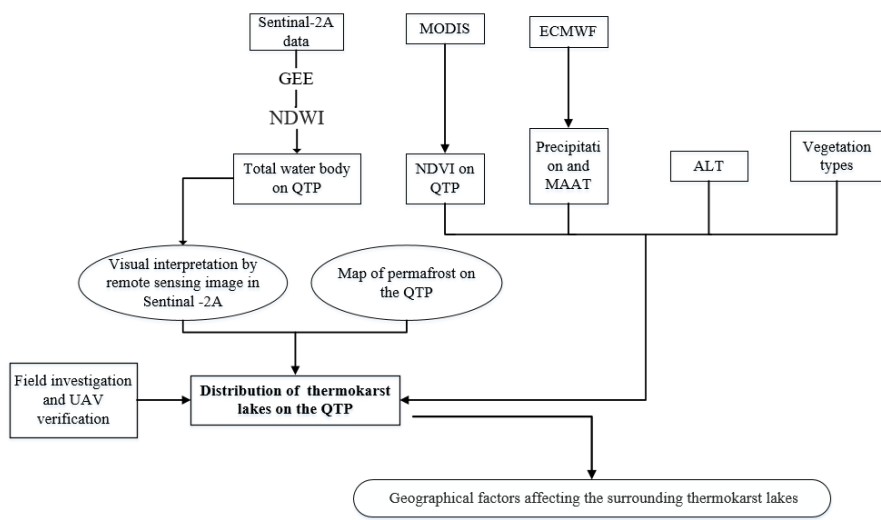

**Figure. 2 Schematic diagram of the process for studying the distribution and influencing factors of**
**thermokarst lakes in the Qinghai-Tibetan Plateau**

**4.2 Thermokarst lakes identification**

When ground ice or ice-rich permafrost thaws, thermokarst lakes or ponds gradually form due to the

surface water accumulation following ground subsidence. Our field investigation showed that more than

90% of lakes along the Qinghai-Tibet Highway have an area of less than 5,000 m², with an average area

of 5,039 m², and the largest thermokarst lake had an area of $4.49 \times 10^5$ m² (Niu et al., 2014). The Sentinel-

2A  imagery is applicable to identify water bodies over 350 m² (Freitas et al., 2019). Therefore, we

assumed that the area of thermokarst lakes in the permafrost regions ranged from 350 to $5.0 \times 10^5$ m².

Although there is a possibility that additional water bodies outside of this area were also thermokarst

lakes, this assumption does represent the most likely thermokarst lake distribution in permafrost regions.


**4.3 GEE processing**

GEE is a geospatial processing platform which utilizes Google's cloud computing resources and large
datasets, making it possible to process, compute, and analyze large and useful data sets from MODIS
data and Sentinel satellite data, as well as climatic and hydrological data, and other reanalysis products
(Gorelick et al., 2017). Through the GEE platform, to automatically extract the total water body



(Sentinel-2A) and environmental factors of the QTP in 2018.

**4.4 Normalized Difference Water Index (NDWI)**

Based on the GEE platform and Sentinel 2A L1C data, the NDWI was used to extract the water bodies
(Mcfeeters and S., 1996).The calculation formula is as follows Eq. (1):

$$NDWI = \frac{GREEN - NIR}{GREEN + NIR} \quad , \tag{1}$$

where GREEN is the green light band and NIR is in the near-infrared band. NDWI is effective in
extracting water information from images by inhibiting vegetation and highlighting water bodies.

Sentinel-2 MSI images include SWIR bands with a resolution of 20 m and green and near-infrared bands
with a resolution of 10 m, making it possible to extract water bodies with a spatial resolution of 10 m.

**4.5 Extraction of water body boundary**

The water index can highlight the difference between the water body and other terrestrial features, while
the threshold value should be established to extract the water body boundary. It has been suggested that
threshold values should be adjusted to achieve the optimal segmentation effect (Lei Ji, 2009; Huang et
al., 2018). In order to determine the optimal threshold, other data such as high-resolution remote sensing
images and field investigative data from the same area can be combined to reduce the errors of water
bodies (Liu et al., 2012). Generally, lakes are always at a relatively stable water level with a flat surface,
and the gradients of lake surfaces are relatively slight. Thermokarst lakes in the QTP are mainly
distributed in high plains or flat low-lying intermontane basins and valleys, where the slopes are less than
3 °(Pan et al., 2014; Qin et al., 2016). The threshold values for a large number of lake sample images
were studied, and it was found that 0.1, as the threshold during water extraction, could more accurately
extract the exact area of the potential lake area (Li and Sheng, 2012). Therefore, the threshold value of
water body index extraction in this study was set to 0.1.

**4.6 Visual interpretation**

Visual interpretation is the process that obtains the information of specific objects from remote sensing
images through direct observation or auxiliary interpretation instruments. Due to the vast area of the QTP,



the water bodies extracted by GEE have considerable errors in the areas where several images overlap.

The complex environmental conditions of the QTP, such as massive clouds, snow cover, and glaciers, make the data process relatively inaccurate. In addition, many lakes and rivers are interconnected, and are thus difficult to separate using automatic methods. Therefore, we used the visual digitization method to create the final thermokarst lake map. Although this is a time-consuming process, especially for such a large area, this method allowed for lake boundary inspection with the highest quality control, and

ensured consistency. Therefore, on the basis of online water body extraction, images of the corresponding year and the same period were downloaded, and the visual interpretation method was used to correct the extraction results and eliminate the influence of rivers. The Sentinel-2A images used in the study comprised more than one hundred scenes over three months of visual interpretation, while the large structural lakes, glacial, and river water bodies, which were automatically extracted by the Google Earth

Engine platform, were accordingly removed, thereby correcting the locations of the thermokarst lakes.

### 4.7 Normalized difference vegetation index (NDVI)

As a good indicator of vegetation activities, NDVI is calculated from infrared bands and near-infrared bands of remote sensing data. The calculation method (Chander et al., 2009) is as follows Eq. (2):

$$NDVI = \frac{NIR - RED}{NIR + RED} , \tag{2}$$

where NIR is the reflectance value of the near-infrared band and RED is the reflectance value of the red band. Based on the GEE platform and Landsat8 L1T data, NDVI data for the QTP in 2018 was extracted. To obtain the NDVI values in a given catchment, we set a buffer zone around the thermokarst lake, and then extracted the NDVI values in the buffer areas, and further calculated the average value in the buffer

zones.

### 4.8 Accuracy verification

The thermokarst lakes along the Qinghai-Tibet Highway were surveyed using an unmanned aerial vehicle (UAV) from September 24 to 28, 2019, and on June 30, 2020. A total of 56 thermokarst lakes were

investigated (Figure. 3): 11 thermokarst lakes of < 1,000 m² 31 thermokarst lakes of 1,000–10,000 m² 10 thermokarst lakes of 10,000–100,000 m² and 4 thermokarst lakes of > 100,000 m² Real-time

kinematic (RTK) positioning sites were also used for correction and accuracy evaluation, and the

calculated mean ground sampling distance (GSD, equivalent to the ground resolution in satellite remote

sensing) was 2.60 cm. The accuracy assessment showed that some relative errors for small thermokarst

lakes were present, while the relative error was close to 0 for large lakes (Table 1).

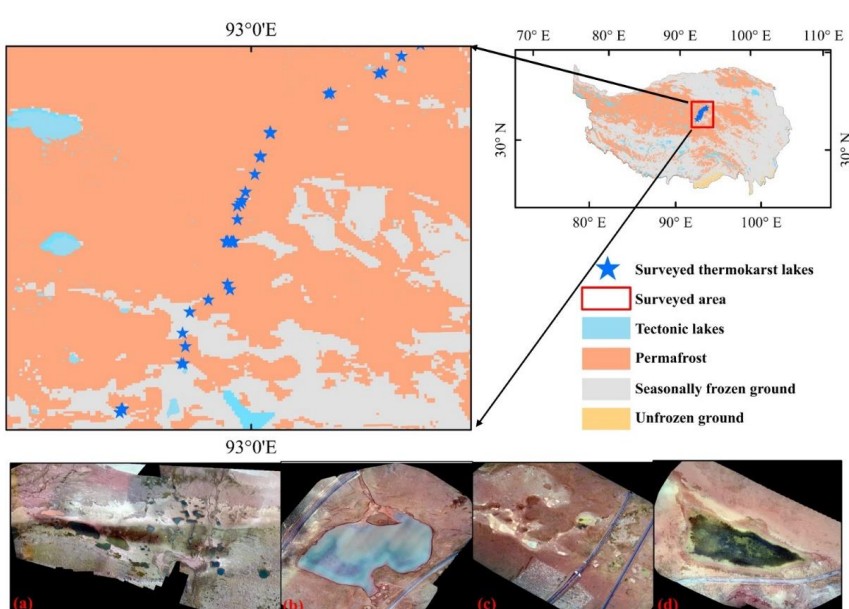

**Figure. 3 Distribution of thermokarst lakes verified by unmanned aerial vehicle (UAV) (a, b, c, and d show**
**examples of four different thermokarst lakes obtained from UAV images)**

**Table 1 Accuracy of thermokarst lakes derived from Sentinel-2 data**

| Time | Site | Longitude | Latitude | Sentinel-2 area (m³) | Field monitoring area (m³) | Relative error |
|---|---|---|---|---|---|---|
| 201307 | TL-1 | 93.516 | 35.390 | 19479.00 | 19500.00 | 0.00 |
| 201307 | TL-2 | 93.644 | 35.465 | 32241.00 | 34000.00 | 0.05 |
| 201307 | TL-3 | 93.334 | 35.336 | 14721.00 | 17000.00 | 0.13 |
| 201307 | TL-4 | 92.906 | 34.830 | 4252.00 | 4200.00 | -0.01 |
| 201307 | TL-5 | 92.928 | 34.829 | 15968.00 | 15000.00 | -0.06 |
| 201307 | TL-6 | 92.084 | 34.834 | 3757.00 | 3600.00 | -0.04 |
| 201909 | TL-7 | 92.468 | 34.250 | 4925.56 | 7089.62 | -0.31 |
| 201909 | TL-8 | 92.466 | 34.251 | 1235.89 | 2175.10 | -0.43 |
| 201909 | TL-9 | 92.720 | 34.407 | 179248.58 | 240211.79 | -0.25 |
| 201909 | TL-10 | 92.903 | 34.823 | 795.25 | 1014.33 | -0.22 |



| 201909 | TL-11 | 92.916 | 34.825 | 6069.82 | 8789.89 | -0.31 |
|---|---|---|---|---|---|---|
| 201909 | TL-12 | 92.899 | 34.825 | 576.80 | 425.97 | 0.35 |
| 201909 | TL-13 | 92.899 | 34.825 | 947.58 | 835.36 | 0.13 |
| 201909 | TL-14 | 92.903 | 34.825 | 1205.20 | 1346.20 | -0.10 |
| 201909 | TL-15 | 92.915 | 34.825 | 522.93 | 453.22 | 0.15 |
| 201909 | TL-16 | 92.901 | 34.825 | 808.91 | 519.78 | 0.56 |
| 201909 | TL-17 | 92.905 | 34.825 | 8604.90 | 7626.72 | 0.13 |
| 201909 | TL-18 | 92.923 | 34.825 | 16797.63 | 15065.05 | 0.12 |
| 201909 | TL-19 | 92.919 | 34.825 | 1451.62 | 1575.28 | -0.08 |
| 201909 | TL-20 | 92.919 | 34.825 | 1919.11 | 1924.36 | 0.00 |
| 201909 | TL-21 | 92.901 | 34.826 | 1445.63 | 1460.56 | -0.01 |
| 201909 | TL-22 | 92.896 | 34.826 | 1790.04 | 1758.67 | 0.02 |
| 201909 | TL-23 | 92.896 | 34.826 | 3469.72 | 3922.01 | -0.12 |
| 201909 | TL-24 | 92.891 | 34.826 | 2226.62 | 1976.09 | 0.13 |
| 201909 | TL-25 | 92.895 | 34.827 | 4426.53 | 5440.25 | -0.19 |
| 201909 | TL-26 | 92.898 | 34.827 | 677.06 | 647.13 | 0.05 |
| 201909 | TL-27 | 92.928 | 34.827 | 2104.42 | 2345.56 | -0.10 |
| 201909 | TL-28 | 92.899 | 34.827 | 795.88 | 798.44 | 0.00 |
| 201909 | TL-29 | 92.900 | 34.827 | 722.73 | 813.95 | -0.11 |
| 201909 | TL-30 | 92.900 | 34.827 | 409.88 | 421.36 | -0.03 |
| 201909 | TL-31 | 92.893 | 34.827 | 1481.51 | 1597.29 | -0.07 |
| 201909 | TL-32 | 92.907 | 34.827 | 1048.39 | 1273.23 | -0.18 |
| 201909 | TL-33 | 92.906 | 34.827 | 1160.40 | 1892.66 | -0.39 |
| 201909 | TL-34 | 92.956 | 34.957 | 31300.09 | 40336.44 | -0.22 |
| 201909 | TL-35 | 93.079 | 35.200 | 399.41 | 426.96 | -0.06 |
| 201909 | TL-36 | 93.079 | 35.201 | 3000.22 | 3360.37 | -0.11 |
| 202006 | TL-37 | 93.622 | 35.453 | 4057.57 | 4304.70 | -0.06 |
| 202006 | TL-38 | 93.620 | 35.452 | 2463.04 | 2923.97 | -0.16 |
| 202006 | TL-39 | 93.445 | 35.366 | 7230.44 | 5447.36 | 0.33 |
| 202006 | TL-40 | 92.455 | 34.238 | 183028.07 | 188379.17 | -0.03 |
| 202006 | TL-41 | 92.464 | 34.250 | 3716.95 | 4012.65 | -0.07 |
| 202006 | TL-42 | 92.465 | 34.250 | 1426.25 | 1534.46 | -0.07 |
| 202006 | TL-43 | 92.466 | 34.251 | 771.51 | 1235.89 | -0.38 |
| 202006 | TL-44 | 92.468 | 34.250 | 4314.42 | 4925.56 | -0.12 |
| 202006 | TL-45 | 92.467 | 34.249 | 1135.53 | 2059.87 | -0.45 |
| 202006 | TL-46 | 92.469 | 34.249 | 3434.33 | 3234.69 | 0.06 |
| 202006 | TL-47 | 92.471 | 34.248 | 1122.13 | 823.95 | 0.36 |
| 202006 | TL-48 | 92.937 | 34.943 | 15491.89 | 13646.70 | 0.14 |
| 202006 | TL-49 | 92.490 | 34.282 | 3227.27 | 2141.44 | 0.51 |
| 202006 | TL-50 | 92.603 | 34.376 | 39462.39 | 33975.36 | 0.16 |
| 202006 | TL-51 | 92.728 | 34.466 | 3389.81 | 3504.69 | -0.03 |
| 202006 | TL-52 | 93.621 | 35.452 | 11658.92 | 7634.86 | 0.53 |
| 202006 | TL-53 | 93.022 | 35.068 | 230441.21 | 217203.89 | 0.06 |
| 202006 | TL-54 | 93.015 | 35.057 | 65969.08 | 71731.06 | -0.08 |





| 202006 | TL-55 | 93.016 | 35.048 | 3756.45 | 4325.55 | -0.13 |
| 202006 | TL-56 | 92.970 | 34.967 | 127724.27 | 156524.70 | -0.18 |

### 5. Distribution of thermokarst lakes

A total of 121151 thermokarst lakes were identified on the QTP, comprising a total thermokarst lake area

of 1730.34 m $km^2$, and accounting for 0.16% of the permafrost area (Figure. 4). The lakes were mainly

distributed in the central and western regions, with an overall density of 12/100 m $km^2$. Thermokarst

lakes less than 1,000 m² accounted for 24% of the total thermokarst lake numbers, while those larger

than 150,000 m² accounted for 50% of the total thermokarst lake area (Figure. 5).

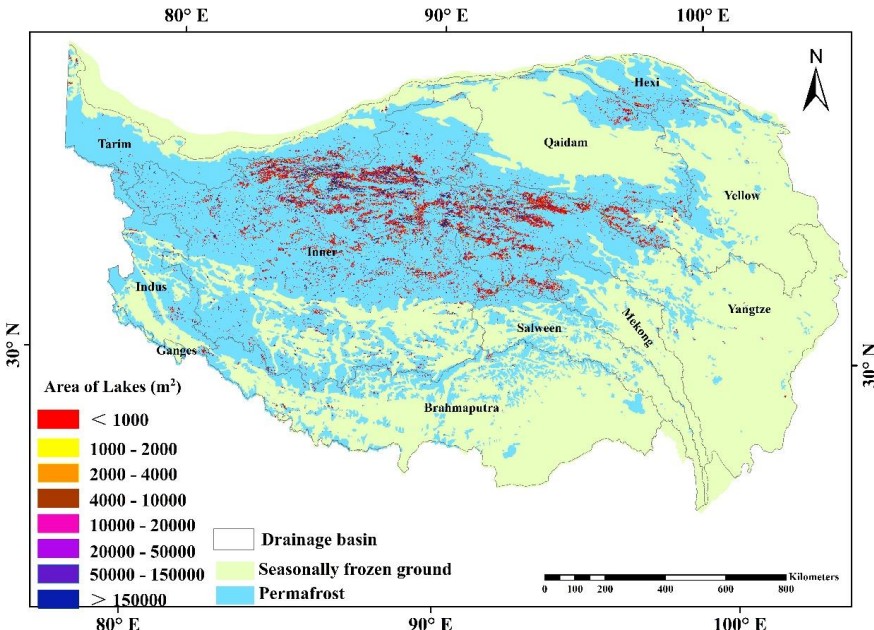


**Figure. 4 Thermokarst lakes in the permafrost regions of the Qinghai-Tibetan Plateau**

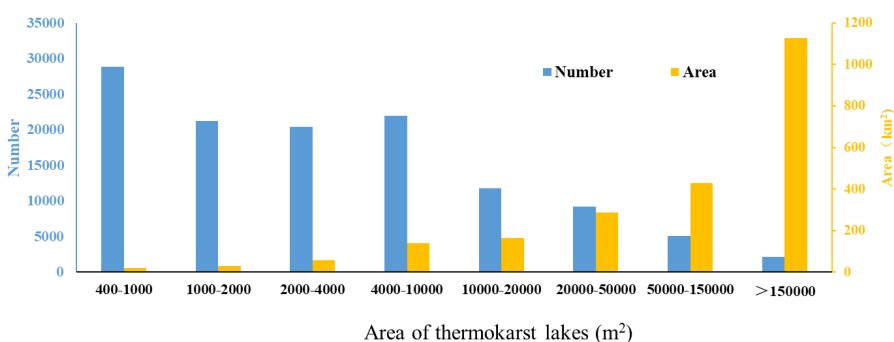

Figure. 5 Number and area of thermokarst lakes on the Qinghai-Tibetan Plateau in 2018


The number and area of thermokarst lakes at altitudes of 5,000 m increased with elevation, and then decreased with elevation, while overall, thermokarst lakes on the QTP were mainly distributed at elevations of 4,750–5,000 m (Figure. 6). We identified 59,314 thermokarst lakes at elevations of 4,750–5,000 m comprising an area of 874.24 m² km², while the thermokarst lakes at 5,000–5,250 and 4,500–

4,750 m comprised areas of 363.5 and 212.3 m² km², respectively.

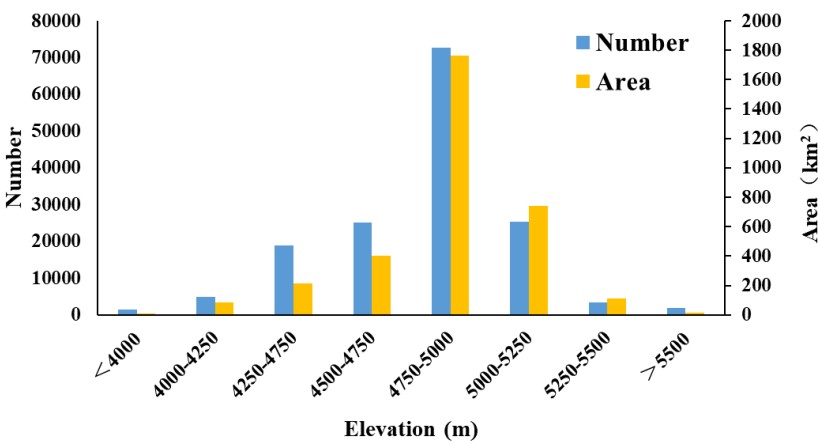

Figure. 6 Number and area of thermokarst lakes at different altitudes on the QTP

The distribution of thermokarst lakes varied greatly among the different vegetation types (Figure. 7). Most thermokarst lakes were distributed in the alpine meadow, steppe, and barren land, while the

associated densities in alpine desert (16/100 km²) and barren land (17/100 km²) were larger than those of

alpine meadows (13/100 km²), alpine steppe (11/100 km²), and wet meadow (11/100 km²). For the

different vegetation types, the percentage of thermokarst lake area under alpine wet meadow was highest,

followed by alpine desert, alpine steppe, and alpine meadow. Barren land had the lowest percentage of

thermokarst lake area.

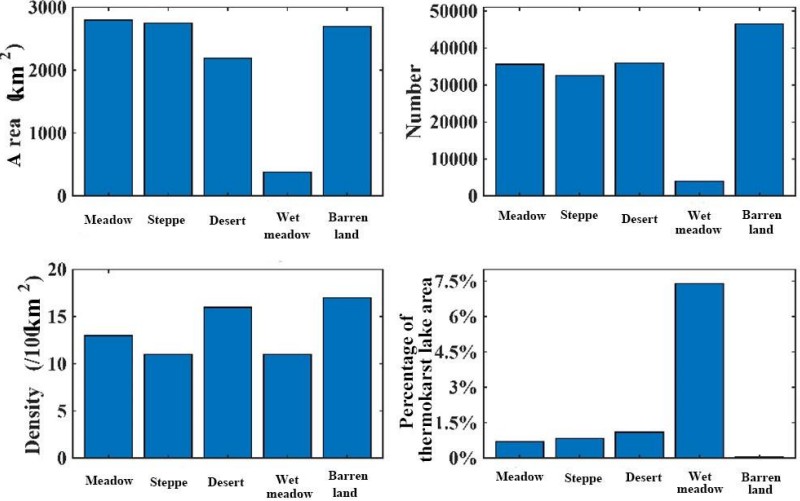

**Figure. 7 Area, number, density, and percentage of thermokarst lakes associated with different vegetation types**


## 6. Relationship between thermokarst lakes and environmental factors

Thermokarst lakes on the QTP were closely related to environmental factors (Figure. 8), as most lakes

were distributed in the areas with an MAGT of -2 to 0 °C and an active layer thickness of 250–300 mm.

According to the distribution of ground ice content, 78% of thermokarst lakes on the QTP were

distributed in areas with ground ice contents higher than 20%. The areas with a mean annual air

temperature of -10 to 5 °C and mean annual precipitation of 400–600 mm had the highest probability of

potentially featuring thermokarst lakes. Soil texture was also associated with thermokarst lakes, as the

highest occurrence of lakes appeared in areas of loamy sand. Although alpine wet meadow featured the



lowest area and number of thermokarst lakes, alpine swamp meadow had the highest probability.

Thermokarst lakes were also related to the NDVI values, as most lakes were distributed in the areas with

NDVI values less than 0.1 (Figure. 9).

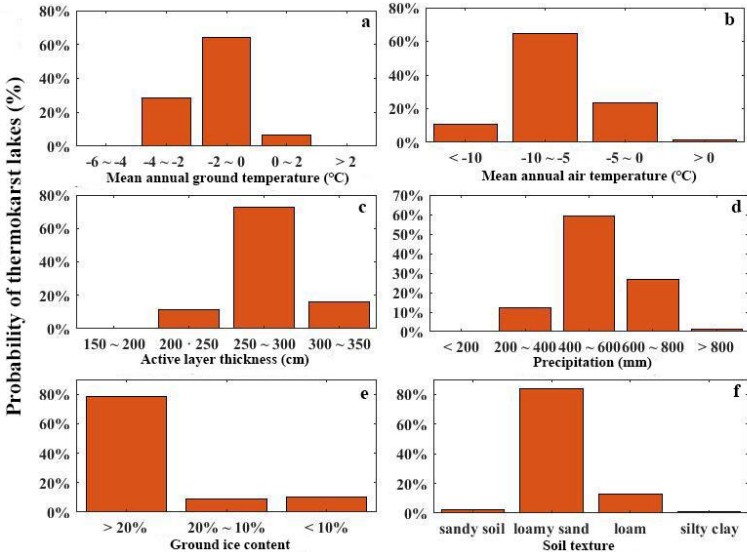

**Figure. 8 Distribution of thermokarst lakes with different mean annual ground temperatures (a); mean**
**annual air temperatures (b); active layer thickness (c); precipitation (d); ground ice content (e); and soil**
**textures (f) on the QTP**

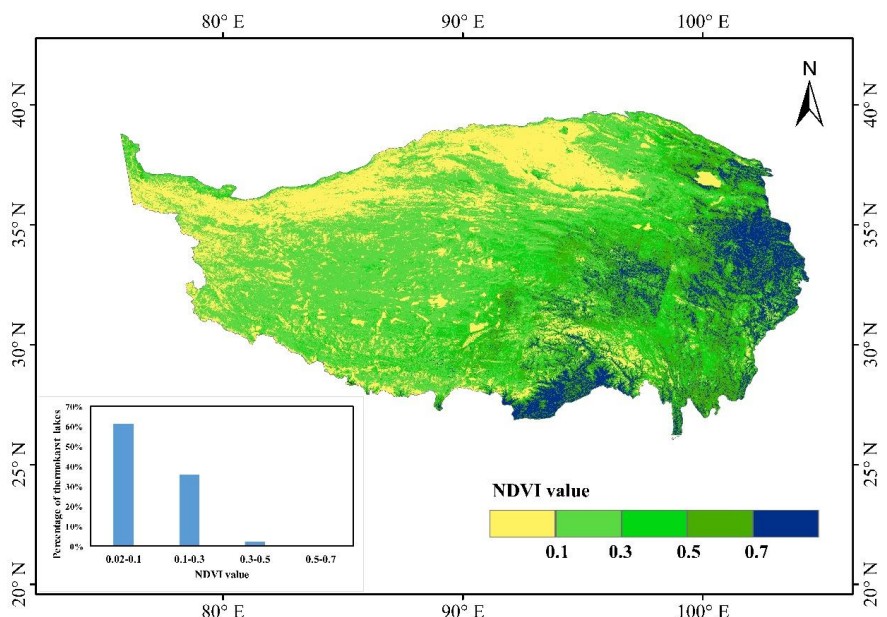

**Figure. 9 Distribution of thermokarst lakes under different NDVI values**


## 7. Comparison and limitations

In general, lakes larger than 1 m㎞²on the QTP have been well documented (Zhang et al., 2014; Wan
et al., 2016; Zhai et al., 2017; Zhang et al., 2017), while fewer reports exist on smaller water bodies.

However, in permafrost regions, omitting small lakes and ponds leads to large underestimations of water
body count and water surface area (Muster et al., 2017). In recent years, the increased availability of
high-resolution satellite imagery, such as the Sentinel-2A satellite, made it possible to study remote
thermokarst lakes in large areas of the unpopulated zone (Kokelf and Jorgenson, 2013). Theoretically,
Sentinel-2 images at a spatial resolution of 10 m can identify ponds smaller than 400 m². In many cases,

ponds smaller than 400 m²can be reliably mapped, and it has been suggested that the Sentinel-2A data
can identify the minimum water body of 350 m²(Freitas et al., 2019). In our study, we extracted water
bodies larger than 400 m²on the QTP, and the UAV multispectral image and ground survey data showed
high accuracy. Notably, the relative error of validation for the thermokarst lakes was related to the

thermokarst lake area, as the smaller lake areas had larger relative errors. This may be explained by the

fact that the small thermokarst lake areas are typically relatively dynamic, showing strong seasonal

changes. Overall, our results showed that the dataset of thermokarst lakes was reliable.

Moreover, our results show that the total area of thermokarst lakes on the QTP is 173,0.34 m km², 

accounting for approximately 0.2% of the permafrost area and 4% of the total water area of lakes and

ponds. The average density of thermokarst lakes was 12/100 m km², which is lower than the circum-

arctic region (28/100 m km²) (Nitze et al., 2018). This can be explained by the fact that the QTP has

lower underground ice content than the Arctic (Mackay, 2015). Thermokarst lakes on the QTP less than

1000 m² accounted for 24% of total thermokarst lake numbers, while it has been found that thermokarst

lakes less than 1,000 m² in the Siberian also accounted for a large proportion of the total lakes (Grosse

et al., 2008). These findings confirm that thermokarst lakes usually have smaller areas. In this study, the

area of thermokarst lakes larger than 150,000 m² accounted for 50% of the total thermokarst lake area,

suggesting that large thermokarst lakes play an important role in carbon and water cycling (Olefeldt et

al., 2016). According to the QTP lake survey data from 2000, the density of lakes over 1 m km² on the

QTP was 1.1/100 m km² (Zhang et al., 2014). Compared with the density of thermokarst lakes in this

study, it can be seen that thermokarst lake density has increased on the QTP (Zhang et al., 2014).

According to our results, thermokarst lakes on the QTP are mainly distributed in places where the mean

annual air temperature ranges from -10 to -5 ℃, and the area of MATGs of -2 to 0 ℃ occupied the most

thermokarst lakes. Previous results have shown that the average active layer thickness on the QTP was

approximately 2.3 m (ranging from 2.2–2.4 m), 80% of which was concentrated in the depth range of

0.8–3.5 m (Qin et al., 2018). Although these factors are associated with the development of thermokarst

areas, it is difficult to draw robust conclusions about the relationships between thermokarst lake areas

and factors related to the area of larger lakes.

Lower permafrost elevations occur largely in the eastern part of the plateau with higher precipitation. For

the middle and western parts of the plateau, permafrost largely exists in areas with elevations higher than

4,000 m (Zou et al., 2017). Areas higher than 5,000 m largely belong to mountain areas, with rugged

topography and steep slopes (Dong et al., 2010), which makes it difficult to form lakes. The vast areas

of QTP are desert steppe and steppe, which have low precipitation and low NDVI values. Therefore, the

thermokarst lakes are mainly distributed in areas with low NDVI values. Instead, the wet meadow, which



has higher NDVI values, accounts for only a small proportion of the vegetative land (4.18%) (Wang et al., 2016). As a result, wet meadows feature the lowest number of thermokarst lakes, yet wet meadows

also feature the highest percentage of thermokarst lakes. This can be explained by the fact that the wet meadows are mainly distributed in the eastern part of the plateau, which usually has an annual precipitation between and 400–600 mm (Gao et al., 2016). Although thermokarst lake areas have not been directly linked to precipitation in western Siberia (Karlsson et al., 2012), it has been suggested that precipitation could affect the total area of thermokarst lakes in Alaska (Swanson, 2019). Considering this

on the QTP, precipitation is the main determinant of lake areas larger than 1 m $km^2$(Zhang et al., 2014), thus it can be seen that thermokarst lakes on the QTP are mostly located in regions with ground ice content higher than 20%. This result is reasonable because the formation of thermokarst lakes is mainly due to ground ice melting (Grosse et al., 2008). In terms of the different soil textures, the highest thermokarst lake probability was found in the loamy sand-type soil. In general on the QTP, the soil

textures are largely characterized by coarse particles (Li et al., 2015), and it should be noted that sandy soils have little probability of thermokarst lakes because such soils are easily eroded or exhibit strong infiltration processes (Wakindiki and Ben-Hur, 2002).

Assessing the impact of climate change on the lake area is of great significance for water resource management and ecological protection (Yuan et al., 2016). In the past 50 years, the average precipitation

on the QTP has shown a slight increasing trend, while the average annual temperature has increased significantly (Zhang et al., 2018). Meanwhile, the NDVI values have increased significantly since the 1980s (Shen et al., 2015). Based on these trends and the formation mechanisms of thermokarst lakes, it could be inferred that thermokarst lakes will likely increase both in numbers and total area in the future, which may greatly affect land surface processes on the QTP.


**8. Data availability**

The dataset developed in this study comprises one.shp file documents containing the thermokarst lake inventory of the QTP region in 2018. The dataset can now be accessed via the National Tibetan Plateau Data Center(Chen et al., 2021) with DOI: 10.11888/Geocry.tpdc.271205. They and can be downloaded

at https://data.tpdc.ac.cn/en/data/c0c05207-568d-41db-ab94-8610bdcdbbe5/.





### 9. Conclusions

In the QTP permafrost regions, approximately 121,000 thermokarst lakes larger than 400 m² were identified, comprising a total lake area of 1730.34 km² and accounting for 0.20% of the total permafrost

area. Most of these thermokarst lakes are smaller than 50,000 m² and are mainly distributed at altitudes of 4,750–5,250 m, with slopes of less than 5 °. These lakes were mainly recorded in areas with an active layer thickness of 250–300 cm, mean annual air temperatures of -10 to 5 ℃, MAGTs of -4 to 0 ℃, and annual precipitation of 400–600 mm. The alpine desert steppe land type was found to feature the largest number of thermokarst lakes, followed by the alpine meadow, while the percentage of thermokarst lakes

was highest in the wet meadow area. Owing to the current technical limitations, it was difficult to investigate thermokarst lakes less than 400 m² in area, thus future work is required to create a dataset that includes these smaller water bodies.

### Acknowledgements.

This work was supported by the National Natural Science Foundation of China (41871050, 41941015),

the National Key Research and Development Program of China (2019YFA0607003), and the Open Foundations of the State Key Laboratory of Frozen Soil Engineering (Grant No. SKLFSE201705).

### Uncategorized References

Brown, J., O.J. Ferrians, Jr., J.A. Heginbottom, and E.S. Melnikov.. 2002. Circum-Arctic Map of Permafrost and Ground-Ice Conditions. Version 2. [indicate subset used]. Boulder.
Chander, G., Markham, B.L., Helder, D.L., 2009. Summary of Current Radiometric Calibration Coefficients for Landsat MSS, TM, ETM+, and EO-1 ALI Sensors. Remote Sensing of Environment 113, 893-903.

Chen, M., Rowland, J.C., Wilson, C.J., Altmann, G.L., Brumby, S.P., 2014. Temporal and spatial pattern of thermokarst lake area changes at Yukon Flats, Alaska. Hydrological Prochydrological Processesesses.
Chen, X., Mu, C., Jia, L., Li, Z., Fan, C., Mu, M., Peng, X., Wu, X., 2021. Thermokarst lakes on the Qinghai-Tibet Plateau (2018). National Tibetan Plateau Data Center.

Cheng, G., Zhao, L., Li, R., Wu, X., Sheng, Y., Hu, G., Zou, D., Jin, H., Li.Xin, Wu, Q., 2019. Characteristics, changes and influences of permafrost on the Qinghai-Tibet Plateau. Chinese Science Bulletin 64, 2783-2795.
Dee, D.P., Uppala, S.M., Simmons, A.J., Berrisford, P., Poli, P., Kobayashi, S., Andrae, U., Balmaseda, M.A., Balsamo, G., Bauer, P., Bechtold, P., Beljaars, A.C.M., van de Berg, L., Bidlot, J.,





Bormann, N., Delsol, C., Dragani, R., Fuentes, M., Geer, A.J., Haimberger, L., Healy, S.B., Hersbach,
        H., Hólm, E.V., Isaksen, L., Kållberg, P., Köhler, M., Matricardi, M., McNally, A.P., Monge-Sanz,
        B.M., Morcrette, J.-J., Park, B.-K., Peubey, C., de Rosnay, P., Tavolato, C., Thépaut, J.-N., Vitart, F.,
        2011. The ERA-Interim reanalysis: configuration and performance of the data assimilation
        system. Quarterly Journal of the Royal Meteorological Society 137, 553-597.

Dong, G., Yi, C., Chen, L., 2010. An introduction to the physical geography of the Qiangtang
        Plateau: A frontier for future geoscience research on the Tibetan Plateau. Physical Geography 31,
        475-492.

        Drusch, M., Bello, U.D., Carlier, S., Colin, O., Fernandez, V., Gascon, F., Hoersch, B., Isola, C.,
        Laberinti, P., Martimort, P., 2012. Sentinel-2: ESA's Optical High-Resolution Mission for GMES

Operational Services. Remote Sensing of Environment 120, 0-36.

        Freitas, P., Vieira, G., Canário, J., Folhas, D., Vincent, W., 2019. Identification of a Threshold
        Minimum Area for Reflectance Retrieval from Thermokarst Lakes and Ponds Using Full-Pixel
        Data from Sentinel-2. Remote Sensing 11.

        Gao, Q., Guo, Y., Xu, H., Ganjurjav, H., Li, Y., Wan, Y., Qin, X., Ma, X., Liu, S., 2016. Climate change

and its impacts on vegetation distribution and net primary productivity of the alpine ecosystem
        in the Qinghai-Tibetan Plateau. Science of the Total Environment 554-555, 34-41.

        Gorelick, N., Hancher, M., Dixon, M., Ilyushchenko, S., Moore, R., 2017. Google Earth Engine:
        Planetary-scale geospatial analysis for everyone. Remote Sensing of Environment.

        Grosse, G., Romanovsky, V., Walter, K., Morgenstern, A., Lantuit, H., Zimov, S., 2008. Distribution

of thermokarst lakes and ponds at three yedoma sites in Siberia.

        Hu, G., Zhao, L., Li, R., Wu, X., Wu, T., Xie, C., Zhu, X., Su, Y., 2019. Variations in soil temperature
        from 1980 to 2015 in permafrost regions on the Qinghai-Tibetan Plateau based on observed
        and reanalysis products. Geoderma 337, 893-905.

        Huang, C., Chen, Y., Zhang, S., Wu, J., 2018. Detecting, extracting and monitoring surface water

from space using optical sensors - a review. Reviews of Geophysics.

        Huang, L., Luo, J., Lin, Z., Niu, F., Liu, L., 2020. Using deep learning to map retrogressive thaw
        slumps in the Beiluhe region (Tibetan Plateau) from CubeSat images. Remote Sensing of
        Environment 237, 111534.

        Jones, B.M., Grosse, G., Arp, C.D., Jones, M.C., Walter Anthony, K.M., Romanovsky, V.E., 2011.

Modern Thermokarst Lake Dynamics in the Continuous Permafrost Zone, Northern Seward
        Peninsula, Alaska. Journal of Geophysical Research Biogeoences 116, G00M03.

        Karlsson, J.M., Lyon, S.W., Destouni, G., 2012. Thermokarst lake, hydrological flow and water
        balance indicators of permafrost change in Western Siberia. Journal of Hydrology 464-465, 459-
        466.

Kokelf, Jorgenson, 2013. Advances in Thermokarst Research. Permafrost & Periglacial Processes
        24, 108-119.

        Kozarski, S., Marks, L., Repelewska-Pękalowa, J., 1998. Multi-Language Glossary of Permafrost
        and Related Ground-Ice Terms.

        Lei Ji, L.Z., and Bruce Wylie, 2009. Analysis of Dynamic Thresholds for the Normalized Difference

Water Index. Photogrammetric Engineering & Remote Sensing 75, 1307-1317.

        Li, J., Roy, D.P., 2017. A Global Analysis of Sentinel-2A, Sentinel-2B and Landsat-8 Data Revisit
        Intervals and Implications for Terrestrial Monitoring. Remote Sensing 9.

        Li, J., Sheng, Y., 2012. An automated scheme for glacial lake dynamics mapping using Landsat



imagery and digital elevation models: a case study in the Himalayas. International Journal of Remote Sensing 33, p.5194-5213.

Li, W., Zhao, L., Wu, X., Wang, S., Sheng, Y., Ping, C., Zhao, Y., Fang, H., Shi, W., 2015. Soil distribution modeling using inductive learning in the eastern part of permafrost regions in Qinghai–Xizang (Tibetan) Plateau. Catena 126, 98-104.

Liu, Y., Chen, H., Zhang, G., Sun, J., Wang, H., 2019. The advanced South Asian monsoon onset accelerates lake expansion over the Tibetan Plateau. ence Bulletin.

Liu, Y., Song, P., Peng, J., Ye, C., 2012. A physical explanation of the variation in threshold for delineating terrestrial water surfaces from multi-temporal images: effects of radiometric correction. International Journal of Remote Sensing 33, 5862-5875.

Luo, J., Niu, F., Lin, Z., Liu, M., Yin, G., 2015. Thermokarst lake changes between 1969 and 2010 in the Beilu River Basin, Qinghai–Tibet Plateau, China. Science Bulletin 60, 556-564.

Luo, J., Niu, F., Lin, Z., Liu, M., Yin, G., 2019. Recent acceleration of thaw slumping in permafrost terrain of Qinghai-Tibet Plateau: An example from the Beiluhe Region. Geomorphology 341, 79-85.

Mackay, J.R., 2015. THE WORLD OF UNDERGROUND ICE. Annals of the Association of American Geographers 62, 1-22.

Marsh, P., Russell, M., Pohl, S., Haywood, H., Onclin, C., 2009. Changes in thaw lake drainage in the Western Canadian Arctic from 1950 to 2000. Hydrological Processes 23, 145-158.

Mcfeeters, S., K., 1996. The use of the Normalized Difference Water Index (NDWI) in the delineation of open water features. International Journal of Remote Sensing 17, 1425-1432.

Morgenstern, A., Grosse, G., Günther, F., Fedorova, I., Schirrmeister, L., 2011. Spatial analyses of thermokarst lakes and basins in Yedoma landscapes of the Lena Delta. The Cryosphere,5,4(2011-10-19) 5, 849-867.

Mu, C., Abbott, B.W., Norris, A.J., Mu, M., Fan, C., Chen, X., Jia, L., Yang, R., Zhang, T., Wang, K., 2020a. The status and stability of permafrost carbon on the Tibetan Plateau. Earth-Science Reviews, 103433.

Mu, C., Shang, J., Zhang, T., Fan, C., Wang, S., Peng, X., Zhong, W., Zhang, F., Mu, M., Jia, L., 2020b. Acceleration of thaw slump during 1997–2017 in the Qilian Mountains of the northern Qinghai-Tibetan plateau. Landslides 17, 1051-1062.

Mu, C., Zhang, T., Wu, Q., Peng, X., Zhang, P., Yang, Y., Hou, Y.W., Zhang, X., Cheng, G., 2016. Dissolved organic carbon, $CO_2$, and $CH_4$ concentrations and their stable isotope ratios in thermokarst lakes on the Qinghai-Tibetan Plateau. J. Limnol. 75, 313-319.

Muster, S., Roth, K., Langer, M., Lange, S., Cresto Aleina, F., Bartsch, A., Morgenstern, A., Grosse, G., Jones, B., Sannel, A.B.K., Sjöberg, Y., Günther, F., Andresen, C., Veremeeva, A., Lindgren, P.R., Bouchard, F., Lara, M.J., Fortier, D., Charbonneau, S., Virtanen, T.A., Hugelius, G., Palmtag, J., Siewert, M.B., Riley, W.J., Koven, C.D., Boike, J., 2017. PeRL: a circum-Arctic Permafrost Region Pond and Lake database. Earth Syst. Sci. Data 9, 317-348.

Nitze, I., Grosse, G., Jones, B.M., Romanovsky, V.E., Boike, J., 2018. Remote sensing quantifies widespread abundance of permafrost region disturbances across the Arctic and Subarctic. Nature Communications 9.

Niu, F., Lin, Z., Hua, L., Lu, J., 2011. Characteristics of thermokarst lakes and their influence on permafrost in Qinghai–Tibet Plateau. Geomorphology 132, 0-233.

Niu, F., Luo, J., Lin, Z., Liu, M., Yin, G., 2014. Morphological Characteristics of Thermokarst Lakes





along the Qinghai-Tibet Engineering Corridor. Arctic Antarctic & Alpine Research 46, 963-974.

Olefeldt, D., Goswami, S., Grosse, G., Hayes, D., Hugelius, G., Kuhry, P., McGuire, A.D.,
Romanovsky, V., Sannel, A.B.K., Schuur, E., 2016. Circumpolar distribution and carbon storage of thermokarst landscapes. Nature communications 7, 1-11.

Ouma, Y.O., Tateishi, R., 2006. A water index for rapid mapping of shoreline changes of five East African Rift Valley lakes: an empirical analysis using Landsat TM and ETM+ data. International Journal of Remote Sensing 27, 3153-3181.

Pan, X., You, Y., Roth, K., Guo, L., Wang, X., Yu, Q., 2014. Mapping Permafrost Features that Influence the Hydrological Processes of a Thermokarst Lake on the Qinghai-Tibet Plateau, China. Permafrost & Periglacial Processes 25, 60-68.

Peng, X., Zhang, T., Frauenfeld, O., Wang, K., Luo, D., Cao, B., Su, H., Jin, H., Wu, Q., 2018. Spatiotemporal Changes in Active Layer Thickness under Contemporary and Projected Climate in
the Northern Hemisphere. Journal of Climate 31, 251-266.

Qin, D., Yao, T., Ding, Y., 2016. Glossary of Cryospheric Science, Beijing: China Meteorological Press.

Qin, y., Tonghua, W.U., Ren, L.I., Guojie, H.U., Yongping, Q., Xiaofan, Z., Shuhua, Y., Wenjun, Y.U., Weihua, W., 2018. Thermal condition of the active layer on the Qinghai-Tibet Plateau simulated
by using the Model of GIPL2. Journal of Glaciology and Geocryology.

Ran, Y., Li, X., Cheng, G., 2018. Climate warming over the past half century has led to thermal degradation of permafrost on the Qinghai–Tibet Plateau. The Cryosphere 12, 595-608.

Ran, Y., Li, X., Cheng, G., Nan, Z., Che, J., Sheng, Y., Wu, Q., Jin, H., Luo, D., Tang, Z., 2020. Mapping the permafrost stability on the Tibetan Plateau for 2005–2015. Science China Earth
Sciences, 1-18.

Reuter, H.I., Nelson, A., Jarvis, A., 2007. An evaluation of void-filling interpolation methods for SRTM data. International Journal of Geographical Information Science 21, 983-1008.

Riordan, B., Verbyla, D., Mcguire, A.D., 2006. Shrinking ponds in subarctic Alaska based on 1950-2002 remotely sensed images. Journal of Geophysical Research Biogeosciences 111, n/a-n/a.

Shen, M., Piao, S., Dorji, T., Liu, Q., Cong, N., Chen, X., An, S., Wang, S., Wang, T., Zhang, G., 2015. Plant phenological responses to climate change on the Tibetan Plateau: research status and challenges. National Science Review 2, 454-467.

Swanson, D.K., 2019. Thermokarst and precipitation drive changes in the area of lakes and ponds in the National Parks of northwestern Alaska, 1984–2018. Arctic, Antarctic, and Alpine Research
530    51, 265-279.

Vonk, J.E., Gustafsson, Ö., 2013. Permafrost-carbon complexities. Nature Geoscience 6, 675-676.

Wakindiki, I.I.C., Ben-Hur, M., 2002. Soil mineralogy and texture effects on crust micromorphology, Infiltration, and Erosion. Soil Science Society of America Journal 66, 897-905.

Wan, W., Long, D., Hong, Y., Ma, Y., Yuan, Y., Xiao, P., Duan, H., Han, Z., Gu, X., 2016. A lake data
set for the Tibetan Plateau from the 1960s, 2005, and 2014. Scientific Data 3.

Wang, Z., Wang, Q., Zhao, L., Wu, X.-d., Yue, G.-y., Zou, D.-f., Nan, Z.-t., Liu, G.-y., Pang, Q.-q., Fang, H.-b., 2016. Mapping the vegetation distribution of the permafrost zone on the Qinghai-Tibet Plateau. Journal of Mountain Science 13, 1035-1046.

Wu, Q., Zhang, P., Jiang, G., Yang, Y., Deng, Y., Wang, X., 2014. Bubble emissions from
thermokarst lakes in the Qinghai–Xizang Plateau. Quaternary International 321, 65-70.

Xu, H., 2006. Modification of normalised difference water index (NDWI) to enhance open water



features in remotely sensed imagery. International Journal of Remote Sensing 27, p.3025‑3033.

Yang, M., Wang, X., Pang, G., Wan, G., Liu, Z., 2019. The Tibetan Plateau cryosphere: Observations and model simulations for current status and recent changes. EARTH ENCE REVIEWS 190, 353‑369.

Yuan, Y., Wan, J., Wan, w., 2016. Remote sensing analysis and Research on lake Environmental changes in the Qinghai‑Tibet Plateau over the past 40 years. Environmental science and management 41, 8‑11.

Zhai, Zhaokun, Xiaosong, Baig, Ali, M.H., Jia, Li, Meng, Jihua, Wei, 2017. Lake water surface mapping in the Tibetan Plateau using the MODIS MOD09Q1 product. Remote sensing letters.

Zhang, Guoqing, Zheng, Guoxiong, Junli, 2017. Lake‑area mapping in the Tibetan Plateau: an evaluation of data and methods. International journal of remote sensing.

Zhang, G., Chen, W., Xie, H., 2019. Tibetan Plateau's lake level and volume changes from NASA's ICESat/ICESat‑2 and Landsat Missions. Geophysical Research Letters 46, 13107‑13118.

Zhang, G., Yao, T., Xie, H., 2014. Lakes' state and abundance across the Tibetan Plateau. Science Bulletin, 3010‑3021.

Zhang, G., Yao, T., Xie, H., Yang, K., Zhu, L., Shum, C.K., Bolch, T., Yi, S., Allen, S., Jiang, L., Chen, W., Ke, C., 2020. Response of Tibetan Plateau lakes to climate change: Trends, patterns, and mechanisms. Earth‑Science Reviews 208, 103269.

Zhang, Z., Chang, J., Xu, C., Zhou, Y., Wu, Y., Chen, X., Jiang, S., Duan, Z., 2018. The response of lake area and vegetation cover variations to climate change over the Qinghai‑Tibetan Plateau during the past 30 years. Science of The Total Environment 635, 443‑451.

Zhao, L., Hu, G., Zou, D., Wu, X., Ma, L., Sun, Z., Yuan, L., Zhou, H., Liu, S., 2019. Permafrost Changes and Its Effects on Hydrological Processes on Qinghai‑Tibet Plateau. . Bulletin of Chinese Academy of Sciences, 1233‑1246.

Zou, D., Lin, Z., Yu, S., Ji, C., Cheng, G., 2017. A New Map of the Permafrost Distribution on the Tibetan Plateau. Chinese Pharmaceutical Affairs 11, 1‑28.