# Peer review of "High-resolution dataset of thermokarst lakes on the Qinghai-Tibetan Plateau"

_Earth System Science Data, 2020_

## Referee Comment (RC1)

*Reviewing process:*

Chen, X., Mu, C., Jia, L., Li, Z., Fan, C., Mu, M., Peng, X., and Wu, X.: High-resolution dataset of thermokarst lakes on the Qinghai-Tibetan Plateau, Earth Syst. Sci. Data Discuss. [preprint], https://doi.org/10.5194/essd-2020-378, in review, 2021.

*General comments:*

The manuscript by Chen et al 2021 tackle an important global hiatus regarding the lacking geographical delineation and monitoring of small waterbodies (< 10 000 m$^2$) in the Earth's surface. The authors focus on the importance of having a better inventory of thermokarst lakes, which are known to be biogeochemical important. The type of research developed by Chen et al 2021 is crucial to support with more complete scientific data to feed Earth System Models and Global Climate Models. To develop the dataset, the authors use Sentinel-2 satellite imagery, taking advantage of its spatial resolution and totally free to use policy. The methodology is not innovative, since it was based on the thresholding definition of a very well know and widely used water index (Normalized Difference Water Index – NDWI) and cloud computing based on Google Earth Engine (GEE), although the results are very interesting when cross-validating with very high resolution satellite imagery, for instance, in Google Earth Pro. The georeferentiation quality seems also good. These results clearly highlight the importance of using Unmanned Aerial Vehicles (UAV) field surveys for better estimate the errors and also the importance of performing intensive visual interpretation of the satellite imagery in order to remove outliers and guarantee the best results. The three months of intensive visual interpretation of more than 100 Sentinel-2 scenes made by the authors is remarkable and was worth it. The simple use of the NDWI for mapping the lakes, using a unique threshold was possible in the Qinghai-Tibetan Plateau (QTP), due to its homogeneous landcover characteristics, but the methodology is not replicable for other permafrost areas, for instance in the boreal forest/tundra zones, were landcover is known to be very heterogeneous. The authors support their work in the work of other authors, using their datasets and trying to provide new advances in the formation and development of thermokarst lakes in the QTP, although the definition of thermokarst lakes that the authors use is very broad and it is probable that some of the lakes that the dataset contains did not have a thermokarst genesis. Field work is still lacking regarding this aspect. To conclude, the dataset created and freely available in a widely used format (shapefile) can be very useful in order to support further methodologies and research, concerning, for instance, small lake expansion/drainage events and the spatial and temporal dynamics of some of its optically active constituents that are known to correlate well with hyperspectral and multispectral data, such as Dissolved Organic Carbon (DOC), Total Suspended Solids (TSS) and others, and with this provide a better understanding on their biogeochemistry role in the thermokarst landscapes of the QTP.

*Specific comments:*

1. Line 24: The authors write: "the true spatial distribution by using a resolution of 10 m with a relative error of 0-0.5", however this a very strong statement. In addition, this relative error, in the way it is, is not easy to understand. How this error was assessed is not clear in the manuscript.
2. Line 38: I suggest eliminating the adjective: "obvious".
3. Line 65: I suggest eliminating the adjective "obvious" to characterize the permafrost degradation of the QTP. Many adjectives in the introduction should be replaced by numerical and quantitative information for a consistent scientific writing.

4. Line 113: Why the authors only mention Sentinel-2A data? Did not the authors use Sentinel-2B data? Why? This is not clear. The authors mention a revisit time for Sentinel-2A of 10 days, but then talk about the twin-satellite system (Sentinel-2A and Sentinel-2B) that have a revisit time of 5 days, without using Sentinel-2B data?

5. Line 122: This sentence is confusing. The Sentinel-2 data is from the European Space Agency (ESA). Note that the Earth Explorer from the United States Geological Survey (USGS) is only an intermediate service to download and access the data. Did not the data was downloaded and accessed in Google Earth Engine? Even in this platform the data provider is the European Union/ESA/Copernicus.

6. Line 135 (figure 1): I would suggest some context information like place labels and also the country boundaries delineation in the small scale map in the upper left corner. Was the map made by the authors or is it from Zou et al (2017)? The subtitle has to be more complete, perhaps mentioning all the sources of information where the auxiliary data came from.

7. Line 138: Why did the authors choose the SRTM of 90 meters instead of the one of 30 meters?

8. Line 140: The end of the sentence is confusing. Which interpolation method are the authors talking about? There are a lot of interpolation methods and this seems a small step of the entire process of generating the SRTM to be highlighted.

9. Line 148: It is not clear why this division was made.

10. Line 172: The authors should support the first sentence of this paragraph with references. See for example: Bouchard, F., MacDonald, L. A., Turner, K. W., Thienpont, J. R., Medeiros, A. S., Biskaborn, B. K., Korosi, J., Hall, R. I., Pienitz, R., & Wolfe, B. B. (2017). Paleolimnology of thermokarst lakes: a window into permafrost landscape evolution. Arctic Science, 3(2), 91–117. https://doi.org/10.1139/as-2016-0022.

11. Line 183: The GEE not only provide MODIS and Sentinel satellite data, but from other satellite constellations also (e.g. Landsat, Sentinel-5 and more). Please, clarify this. The next sentence (beginning at the line 185) has also to be reformulated. Some words are missing.

12. Line 189: Why did the authors use Sentinel-2A L1C data, instead of L2A? Were the images atmospherically corrected and compensated? Did the authors apply the Bidirectional Reflectance Distribution Function (BRDF) and took as reference a Digital Terrain Model to eliminate topographic shadows, for instance in mountain areas? How did the authors manage to solve this type of problems? When visual inspecting the authors database I was able to find some lakes in the mountain areas that were in fact mountain shadows (e.g. Northwest of Tarim). The authors are asked to gently eliminate this type of features and artifacts from the database.

13. Line 192: What do the authors mean with "green light band"? The authors should use references to demonstrate NDWI effectiveness and better supporting its theoretical and physical basis.

14. Line 194: The authors mention the resolution of Sentinel-2 bands previously. This, in the way it is, seems a repetition. Plus, the SWIR bands are not used in this index.

15. Line 215: Why did not the authors use the FMask algorithm (or similar) to previously remove some of this noise of the images, such as clouds, snow and clouds shadow?

16. Line 232: Was the NDVI data extracted from MODIS or Landsat 8? This is not consistent with what the authors mention at the line 136 and further in figure 2.

17. Line 239: Which UAV was used? How was the UAV data processed? Which technique was used? What were the Root-Mean-Square (RSM – X, Y and possibly Z) errors of the generated orthomosacis? These are important details to mention, specially when studying waterbodies due to the lack of contrasting features that make unfeasible the use of some techniques, such as Structure from Motion (SfM). The authors are gently asked to provide new advances on this topic.

18. Line 251 (table 1): How was the relative error calculated? What does it mean? Does the error have units? This is not clear throughout the entire manuscript.

19. Line 260 (figure 4): Although the distribution of thermokarst lakes in the QTP is very interesting, in this map it is not possible to discriminate and understand that distribution. I suggest the authors using a generalization procedure or less classes in order to highlight better all the features (e.g. polygon to point just for visualization purposes in this map or other type of generalization).
20. Line 369: The authors are asked to add a reference in this last statement since some authors also have demonstrated this. See for example: Turetsky, M. R., Abbott, B. W., Jones, M. C., Anthony, K. W., Olefeldt, D., Schuur, E. A. G., Grosse, G., Kuhry, P., Hugelius, G., Koven, C., Lawrence, D. M., Gibson, C., Sannel, A. B. K., & McGuire, A. D. (2020). Carbon release through abrupt permafrost thaw. Nature Geoscience, 13(2), 138–143. https://doi.org/10.1038/s41561-019-0526-0.

*Technical corrections:*

- Sometimes the English is confusing to the reader and I suggest the authors to fully rewrite some of the sentences and ensure that they are clear for the reader.
- The authors used to write as units $m^2km^2$ in many circumstances, but this wrong. The authors should uniformize the working units and fix these issues throughout the entire manuscript.
- If the authors choose to use acronyms, they should use it all the way throughout the manuscript.
- Line 169 (figure 2): Sentinel-2A is not well written in both situations. Since the authors add the source of information in the first row of the figure, I would suggest adding the SRTM right above ALT, and change this last name for topography to be easier to understand.
- Line 220: Online waterbody extraction? What does this mean?

Thank you.

---

## Referee Comment (RC2)

Review of the manuscript
"High-resolution dataset of thermokarst lakes on the Qinghai-Tibetan Plateau"
*Xu Chen, Cuicui Mu, Lin Jia, Zhilong Li, Chengyan Fan, Mei Mu, Xiaoqing Peng, Xiaodong Wu*

The manuscript is devoted to topical issues of determining the number and total area of thermokarst lakes (TL) in the permafrost zone of the Qinghai-Tibet Plateau (QTP). Considering the vastness of the territory and the large number of TLs of different sizes, the determination of their area from remote sensing data with a relatively high spatial resolution (10 m) in itself is a serious methodological problem. The authors provided new experimental data on the amount and area of TL within the QTP, as well as interesting information on the relationship of TL with altitude levels, types of vegetation cover, and environmental factors. At the same time, there are a number of questions about the manuscript, mainly of a methodological nature, which require clarification.

Questions and comments

1. How did thermokarst lakes differ from other types of lakes in the permafrost zone?
2. What does 10a mean in parameter 19.5 cm / 10a, associated with the rate of increase in the active layer thickness (line 100)?
3. It is known that the area of TL, as a rule, having an insignificant depth, significantly depends on the period of floods and precipitation that fell on the eve of the survey. Differences in the area of TL during floods and after it can reach 40%. In this regard, it is necessary to clarify - what are the dates of the space borne survey, the dates of the descent of flood waters and the dates of the passage of precipitation?
4. Explain in comparison with what data the threshold NDWI = 0.1 was set?
5. How and with what data was visual interpretation carried out to ensure "lake boundary inspection with the highest quality control and consistency" (218)? From which resources and which images of bodies of water were downloaded from online for visual interpretation (220)?
6. What does three months of visual interpretation mean (223)?
7. The data obtained show that even for large TLs, the errors are large (up to 20%) (Table 1), which indicates either the low accuracy of the method or the instability of the TL area, including large ones. For comparison with other results and errors in determining the area of TL according to remote sensing data, the authors are recommended to build a graph of the dependence of the modulus of the average and maximum error (or RMSE) on the area of water bodies for groups, for example, for groups of 400-1000 m, 1000-2000 m ... etc.
8. The information about the errors (table 1) is in no way connected with the final result - the estimate of the total area of TLs in the QTP area. In this regard, it is not clear why table 1 is needed? At the same time, the information in table 1 can be used to assess the error in determining the total area of water bodies in the region. Any quantitative estimates are always associated with errors.
9. What information about the number and total area of thermokarst lakes in the region of the entire QTP was previously obtained by other authors, for example, topographic maps of different scales? If there are such data (maps), then it is necessary to show what are the advantages of the data obtained by the authors in comparison with the known data (maps).
10. In fig. 6 some data do not match the description in the text (268, 269, 270). So in the text for an altitude of 4750-5000 m, the number of TLs is 59, 314 and the area they occupy is 874.24 $km^2$. On the graph for this height, the number of TLs is more than 70,000, and the area is about 1800 $km^2$.
11. The map in Fig. 9 does not coincide in contours with the map in Fig. 4.

The manuscript can be published after serious revision.

---

## Author Comment (AC1)

**Detailed responses for the comments**

**General comments:**

The manuscript by Chen et al 2021 tackle an important global hiatus regarding the lacking geographical delineation and monitoring of small waterbodies ($< 10\,000$ m$^2$) in the Earth's surface. The authors focus on the importance of having a better inventory of thermokarst lakes, which are known to be biogeochemical important. The type of research developed by Chen et al 2021 is crucial to support with more complete scientific data to feed Earth System Models and Global Climate Models. To develop the dataset, the authors use Sentinel-2 satellite imagery, taking advantage of its spatial resolution and totally free to use policy. The methodology is not innovative, since it was based on the thresholding definition of a very well know and widely used water index (Normalized Difference Water Index – NDWI) and cloud computing based on Google Earth Engine (GEE), although the results are very interesting when cross-validating with very high resolution satellite imagery, for instance, in Google Earth Pro. The georeferentiation quality seems also good. These results clearly highlight the importance of using Unmanned Aerial Vehicles (UAV) field surveys for better estimate the errors and also the importance of performing intensive visual interpretation of the satellite imagery in order to remove outliers and guarantee the best results. The three months of intensive visual interpretation of more than 100 Sentinel-2 scenes made by the authors is remarkable and was worth it. The simple use of the NDWI for mapping the lakes, using a unique threshold was possible in the Qinghai-Tibetan Plateau (QTP), due to its homogeneous landcover characteristics, but the methodology is not replicable for other permafrost areas, for instance in the boreal forest/tundra zones, were landcover is known to be very heterogeneous. The authors support their work in the work of other authors, using their datasets and trying to provide new advances in the formation and development of thermokarst lakes in the QTP, although the definition of thermokarstlakes that the authors use is very broad and it is probable that some of the lakes that the dataset contains did not have a thermokarst genesis. Field work is still lacking regarding this aspect. To conclude, the dataset created and freely available in a widely used format (shapefile) can be very useful in order to support further

methodologies and research,concerning, for instance, small lake expansion/drainage events and the spatial and temporal dynamics of some of its optically active constituents that are known to correlate well with hyperspectral and multispectral data, such as Dissolved Organic Carbon (DOC), Total Suspended Solids (TSS) and others, and with this provide a better understanding on their biogeochemistry role in the thermokarst landscapes of the QTP.

**Response:** Thank you for your comments for our work.

**Specific comments:**

1. Line 24: The authors write: "the true spatial distribution by using a resolution of 10 m with a relative error of 0-0.5", however this a very strong statement. In addition, this relative error, in the way it is, is not easy to understand. How this error was assessed is not clear in the manuscript.

**Response:** We calculated the coefficient of determination ($R^2$), average absolute error (MAE) and root-mean-square error (RMSE) of the data (Table 1). The MEA and RASE values were close to 1:1, indicating the results were reliable. We also added the calculation method in the revised version.

2. Line 38: I suggest eliminating the adjective: "obvious".

**Response:** Changed. It has been revised as follows: "One of the characteristics of permafrost degradation is the formation of thermokarst terrains.".

3. Line 65: I suggest eliminating the adjective "obvious" to characterize the permafrost degradation of the QTP. Many adjectives in the introduction should be replaced by numerical and quantitative information for a consistent scientific writing.

**Response:** Thanks. It has been revised as follows: "In recent decades, permafrost on the QTP has experienced significant degradation, as is indicated by the increasing ground temperatures.".

4. Line 113: Why the authors only mention Sentinel-2A data? Did not the authors use

Sentinel-2B data? Why? This is not clear. The authors mention a revisit time for Sentinel-2A of 10 days, but then talk about the twin-satellite system (Sentinel-2A and Sentinel-2B) that have a revisit time of 5 days, without using Sentinel-2B data?

**Response:** We did not use Sentinel-2B data because some of these data were not available. To avoid any confusions, we revised these sentence as follows: "The Sentinel-2 mission, organized by the Global Environment and Security Monitoring (GMES), uses a twin-satellite system to capture multi-spectral high-resolution optical observations at high revisit frequencies around the world. Sentinel-2B and Sentinel-2A were launched by the European Space Agency (ESA) on March 7, 2017 and June 23, 2015, respectively. In this study, we used the data of Sentinel-2A because some images of Sentinel-2B were not available on the QTP in 2018.".

5. Line 122: This sentence is confusing. The Sentinel-2 data is from the European Space Agency (ESA). Note that the Earth Explorer from the United States Geological Survey (USGS) is only an intermediate service to download and access the data. Did not the data was downloaded and accessed in Google Earth Engine? Even in this platform the data provider is the European Union/ESA/Copernicus.

**Response:** Thanks for pointing this out. We revised these sentence as follows: "Since December 2015, data can be acquired through free download from the ESA official website (https://scihub.copernicus.eu/). We used the data at 10-m resolution in this study.".

6. Line 135 (figure 1): I would suggest some context information like place labels and also the country boundaries delineation in the small scale map in the upper left corner. Was the map made by the authors or is it from Zou et al (2017)? The subtitle has to be more complete, perhaps mentioning all the sources of information where the auxiliary data came from.

**Response:** In this figure, we added the regional map of southeast asia and the location of field monitoring thermokarst lakes. The subtitle has been changed as follows: "Distribution of permafrost on the Qinghai-Tibetan Plateau (QTP). The permafrost

distribution data is from Zou et al., (2017), and the red star presents the field monitoring thermokarst lakes".

[Figure]

Figure 1: Distribution of permafrost on the Qinghai-Tibetan Plateau (QTP). The permafrost distribution data are from Zou et al., (2017), and the red star presents the field monitoring thermokarst lakes.

7. Line 138: Why did the authors choose the SRTM of 90 meters instead of the one of 30 meters?

**Response:** Presently, the accuracy verification for SRTM of 30 m on the QTP is insufficient due to the complex topography in the mountain permafrost regions. The SRTM of 90 m has been commonly used in China, especially in permafrost regions. Therefore, in this study, SRTM of 90 m was used to study the altitude of the thermokarst lakes. To be clear, we explained this in the revised version as follows: "Presently, it is insufficient for the accuracy verification for SRTM of 30 m on the QTP due to the complex topography, thus SRTM of 90 m were used in this study (Reuter et al., 2007; Global Land Cover, 2018; Li et al., 2016)."

8. Line 140: The end of the sentence is confusing. Which interpolation method are the authors talking about? There are a lot of interpolation methods and this seems a small step of the entire process of generating the SRTM to be highlighted.

**Response:** The SRTM terrain data were from Global Land Cover, 2018, which is cited as Reuter et al. (2007). It has been revised as follows: "The Digital Elevation Model (DEM) dataset with a resolution of 90 m was retrieved from the Shuttle Radar Topography Mission (SRTM) terrain data (Reuter et al., 2007; Global Land Cover, 2018 Li et al., 2016).".

9. Line 148: It is not clear why this division was made.

**Response:** We made these division according to the ground ice content. It was explained as follows: "Ground ice data were retrieved from the map of permafrost and ground ice in the Northern Hemisphere (Brown, 2002), of which the ground ice content in the top 20 m is divided into > 20%, 10–20% and < 10% with the percentage of ice volume".

10. Line 172: The authors should support the first sentence of this paragraph with references. See for example: Bouchard, F., MacDonald, L. A., Turner, K. W., Thienpont, J. R., Medeiros, A.S., Biskaborn, B. K., Korosi, J., Hall, R. I., Pienitz, R., & Wolfe, B. B. (2017). Paleolimnology of thermokarst lakes: a window into permafrost landscape evolution. Arctic Science, 3(2), 91–117. https://doi.org/10.1139/as-2016-0022.

**Response:** Thanks, these references were added.

11. Line 183: The GEE not only provide MODIS and Sentinel satellite data, but from other satellite constellations also (e.g. Landsat, Sentinel-5 and more). Please, clarify this. The next sentence (beginning at the line 185) has also to be reformulated. Some words are missing.

**Response:** These sentences have been revised as follows: "GEE is a geospatial processing platform which utilizes Google's cloud computing resources and large

datasets, making it possible to process, compute, and analyze large data sets from MODIS data, satellite data and other reanalysis products (Gorelick et al., 2017). In this study, the water bodies and environmental factors in 2018 on the QTP can be automatically extracted through the GEE platform.".

12. Line 189: Why did the authors use Sentinel-2A L1C data, instead of L2A? Were the images atmospherically corrected and compensated? Did the authors apply the Bidirectional Reflectance Distribution Function (BRDF) and took as reference a Digital Terrain Model to eliminate topographic shadows, for instance in mountain areas? How did the authors manage to solve this type of problems? When visual inspecting the authors database I was able to find some lakes in the mountain areas that were in fact mountain shadows (e.g. Northwest of Tarim). The authors are asked to gently eliminate this type of features and artifacts from the database.

**Response:** Thanks for this comment. The data before December 2018 only have L1C products, not L2A products of this level, thus we used Sentinel-2A L1C data. In this study, only the GREEN and NIR bands of the Sentinel 2A satellite were used in the automatic extraction. The NDWI was calculated and water body information was extracted in GEE. Thermokarst lakes are usually located in areas with flat topography, where the slopes are less than 3° (Pan et al., 2014; Qin et al., 2016), thus the most mountain shadows can be removed during automatic extraction. In addition, after automatic extraction, mountain shadows were eliminated through the visual interpretation and the location of the thermokarst lakes was corrected. This method was also used in the Wang, X., et al. 2020. Glacial lake inventory of high-mountain Asia in 1990 and 2018 derived from Landsat images. Earth Syst. Sci. Data, 12, 2169–2182.

13. Line 192: What do the authors mean with "green light band"? The authors should use references to demonstrate NDWI effectiveness and better supporting its theoretical and physical basis.

**Response:** Thanks for your suggestions. It has been revised as follows: "The Normalized Difference Water Index (NDWI) has been developed to delineate open

water features and enhance their presence in remotely-sensed digital imagery (Yang et al., 2017). The NDWI makes use of reflected near-infrared radiation and visible green light to enhance the presence of such features while eliminating the presence of soil and terrestrial vegetation features(Gao, 1996, Mcfeeters and S., 1996). NDWI is a useful index to extract water information from images by inhibiting vegetation and highlighting water bodies(Xu, 2006). Based on the GEE platform and Sentinel-2A data, the NDWI was used to extract the water bodies".

14. Line 194: The authors mention the resolution of Sentinel-2 bands previously. This, in the way it is, seems a repetition. Plus, the SWIR bands are not used in this index.

**Response:** Thanks, we simply deleted this sentence.

15. Line 215: Why did not the authors use the FMask algorithm (or similar) to previously remove some of this noise of the images, such as clouds, snow and clouds shadow?

**Response:** To obtain the high-quality data with less noise such as clouds, snow and clouds shadow, we downloaded images from cloudless days between April 1 and October 30, 2018, thus this study need not to remove these noises. In the revised version, we clearly explained this as follows: "To obtain the high-quality data with less noise such as clouds, snow and clouds shadow, we downloaded images from cloudless days between April 1 and October 30, 2018."

16. Line 232: Was the NDVI data extracted from MODIS or Landsat 8? This is not consistent with what the authors mention at the line 136 and further in figure 2.

**Response:** Sorry for the confusion. The NDVI data were extracted from Landsat 8. The sentence in L136 and figure has been revised accordingly.

17. Line 239: Which UAV was used? How was the UAV data processed? Which technique was used? What were the Root-Mean-Square (RSM – X, Y and possibly Z) errors of the generated or thomosacis? These are important details to mention, specially

when studying waterbodies due to the lack of contrasting features that make unfeasible the use of some techniques, such as Structure from Motion (SfM). The authors are gently asked to provide new advances on this topic.

**Response:** Thanks. These information has been added as follows: "The thermokarst lakes along the Qinghai-Tibet Highway were surveyed using an unmanned aerial vehicle (UAV) (DJI PHANTOM 3 4K) from September 24 to 28, 2019, and on June 30, 2020. The UAV data were converted to Digital Orthophoto Map using Pix4Dmapper. The absolute errors of X, Y, Z were between 1.06-1.14 m and root-mean-square (RMS) had a range of 0.46-0.54 m for all project. Meanwhile, the boundary of thermokarst lake was clearly indicated by the water and land, and thus the boundary feature can be obtained.

18. Line 251 (table 1): How was the relative error calculated? What does it mean? Does the error have units? This is not clear throughout the entire manuscript.

**Response:** In the present version, we provided the detailed information of these calculations as follows: "We calculated the $R^2$, MAE and RASE, which were summarized in Table 1. It has been revised as follows: "We used the coefficient of determination ($R^2$), average absolute error (MAE) and root-mean-square error (RMSE) to assess the accuracy. The calculation methods (Draper and Smith, 1998) are as follows Eq. (3-5)."

$$R^2 = 1 - \frac{\sum_i (y_i - f_i)^2}{\sum_i (y_i - \hat{y})^2} \tag{3}$$

$$RMSE = \sqrt{\frac{1}{N} \sum_{i=1}^{N} (Z_{oi} - Z_{pi})^2} \tag{4}$$

$$MAE = \frac{1}{N} \sum_{i=1}^{N} \left| Z_{oi} - Z_{pi} \right| \tag{5}$$

where $y_i$ is the extracted area of Sentinel-2A thermokarst lakes, $f_i$ is the measured value of UAV, $Z_{oi}$ is the measured value of UAV in the $i$ classification, $Z_{pi}$ is the extracted value of Sentinel-2A, and $N$ is the number of lakes in each classification.

19. Line 260 (figure 4): Although the distribution of thermokarst lakes in the QTP is

very interesting, in this map it is not possible to discriminate and understand that distribution. I suggest the authors using a generalization procedure or less classes in order to highlight better all the features (e.g. polygon to point just for visualization purposes in this map or other type of generalization).

**Response:** Thank you for your suggestion. We agree that it not possible to discriminate the distribution of thermokarst lakes in the present figure. However, the most thermokarst lakes are small and the areas varied considerably and it is difficult to use generalization procedures to highlight their features. We already submitted the Shp file to the data source, and thus we believe potential readers can obtain these data.

To show the possible relationship between thermokarst lake and environmental factors, we used the five classes (<1,000 m$^2$, 1,000-10,000 m$^2$, 10,000-50,000 m$^2$, 50,000-150,000 m$^2$, >150,000 m$^2$) to show the distribution of thermokarst lakes.

[Figure]

Figure 4: Distribution of thermokarst lakes in the permafrost regions of the Qinghai-Tibetan Plateau

20. Line 369: The authors are asked to add a reference in this last statement since some authors also have demonstrated this. See for example: Turetsky, M. R., Abbott, B. W.,

Jones, M. C.,Anthony, K. W., Olefeldt, D., Schuur, E. A. G., Grosse, G., Kuhry, P., Hugelius, G., Koven, C.,Lawrence, D. M., Gibson, C., Sannel, A. B. K., & McGuire, A. D. (2020). Carbon release throughabrupt permafrost thaw. Nature Geoscience, 13(2), 138–143.https://doi.org/10.1038/s41561-019-0526-0.

**Response:** Added.

**Technical corrections:**

• Sometimes the English is confusing to the reader and I suggest the authors to fully rewrite some of the sentences and ensure that they are clear for the reader.

**Response:** Thanks, the English has been polished using the AJE company.

**Invoice**

[Figure]

ELSEVIER

| Invoice | |
|---|---|
| Reference: | CS114126 |
| Order nr: | 20146 |
| Date: | Dec 01, 2020 |
| Currency: | USD |

**Product Information**

| Quantity | Description | Unit price | VAT rate | VAT amount | Total amount (usd) | Paid (usd) | Due |
|---|---|---|---|---|---|---|---|
| 1 | Language editing express | 399.00 | 0.0% | 0.00 | 399.00 | 399.00 | 0.00 |
| | | | | | 399.00 | 399.00 | 0.00 |

**Customer Information**

| | |
|---|---|
| Name: | xu chen |
| University: | Lanzhou University |
| Department: | College of Earth and Environmental Sciences |
| VAT Number: | 12100000438001702R |
| Address: | Lanzhou University, Lanzhou |
| Postal code / zip: | 730000 |
| City: | Lanzhou |
| State / province: | |
| Country: | China |

Total VAT amount: USD 0.00

VAT registration numbers: Austria (AT) U62029744, Belgium (BE) 454069965, Canada (CA) 899471825RT0001, Cyprus (CY) 99200008T, Czech Republic (CZ) 680459405, Denmark (DK) 17105779, Estonia (EE) 101100676, Finland (FI) 10123303, France (FR) 67390585230, Germany (DE) 172046177, Greece (GR) 999838602, Hungary (HU) 26952811, Ireland (IE) 9507113Q, Italy (IT) 00119309995, Luxembourg (LU) 21424724, Malta (MT) 18116507, Netherlands (NL) 8014814247B01, Poland (PL) 5262786955, Portugal (PT) 980081963, Slovakia (SK) 4020110281, Slovenia (SI) 55336051, Spain (ES) A0063646D, Sweden (SE) 502048821801, Switzerland (CH) 494663, United Kingdom (GB) 494627212

Paid in Full

• The authors used to write as units m2km2 in many circumstances, but this wrong.

The authors should uniformize the working units and fix these issues throughout the entire manuscript.

**Response:** Changed.

• If the authors choose to use acronyms, they should use it all the way throughout the manuscript.

**Response:** Changed.

• Line 169 (figure 2): Sentinel-2A is not well written in both situations. Since the authors add the source of information in the first row of the figure, I would suggest adding the SRTM right above ALT, and change this last name for topography to be easier to understand.

**Response:** Changed. In this version, the figure is modified as follows:

[Figure]

Figure 2: Schematic diagram of the process for studying the distribution and influencing factors of thermokarst lakes on the Qinghai-Tibetan Plateau

• Line 220: Online waterbody extraction? What does this mean?

**Response:** It has been revised as follows: "on the basis of water body automatic

extraction".

---

## Author Comment (AC2)

**Detailed responses for the comments**

The manuscript is devoted to topical issues of determining the number and total area of thermokarst lakes (TL) in the permafrost zone of the Qinghai-Tibet Plateau (QTP). Considering the vastness of the territory and the large number of TLs of different sizes, the determination of their area from remote sensing data with a relatively high spatial resolution (10 m) in itself is a serious methodological problem. The authors provided new experimental data on the amount and area of TL within the QTP, as well as interesting information on the relationship of TL with altitude levels, types of vegetation cover, and environmental factors. At the same time, there are a number of questions about the manuscript, mainly of a methodological nature, which require clarification.

**Response:** Thank you very much for your detailed suggestion. We revised the manuscript according the reviewers' comments, and we believe the quality of the revised version has been greatly improved.

1. How did thermokarst lakes differ from other types of lakes in the permafrost zone?

**Response:** The formation of thermokarst lakes is due to the surface water accumulation following ground subsidence during permafrost degradation. There are no strict criteria to distinguish the thermokarst lakes and non-thermokarst lakes. However, it is clearly that some tectonic lakes were not thermokarst lakes, which is characterized by large areas. In addition, more than 90% of lakes along the Qinghai-Tibet Highway have an area of less than 5,000 m², with an average area of 5,039 m², and the largest thermokarst lake had an area of $4.49 \times 10^5$ m² (Niu et al., 2014). Thus, the area of thermokarst lakes in this study ranged from 350 m² to $5.0 \times 10^5$ m² and the tectonic lakes are excluded. Although there is a possibility that some of the lakes were formed without thermokarst genesis, this is the first dataset for the thermokarst lakes on the QTP.

These information has been added in the revised version in the Comparison and limitations section as follows:

In our study, we classified that thermokarst lakes according to the area (350 m² to $5.0 \times 10^5$ m²) because the smaller ponds are likely seasonal water bodies and they are also can not be recognized from remote sensing data, while the larger lakes are maybe

tectonic lakes. However, due to there are no strict criteria to distinguish the thermokarst and non-thermokarst lakes, there is a possibility that some larger thermokarst lakes on the QTP were not included in our dataset, future studies pertained to these large lakes should paid more attention on the areas of these lakes.

2. What does 10a mean in parameter 19.5 cm / 10a, associated with the rate of increase in the active layer thickness (line 100)?

**Response:** This sentence has been rewritten as follows: "the active layer thickness along the Qinghai-Tibet Highway has increased at the rate of 19.5 cm/10a from 1982 to 2018".

3. It is known that the area of TL, as a rule, having an insignificant depth, significantly depends on the period of floods and precipitation that fell on the eve of the survey. Differences in the area of TL during floods and after it can reach 40%. In this regard, it is necessary to clarify - what are the dates of the space borne survey, the dates of the descent of flood waters and the dates of the passage of precipitation?

**Response:** Thanks for the review. The dates of the descent of flood waters and passage of precipitation is important for the area of thermokarst lakes. In this study, Sentinel-2A images on cloudless days during April 1 to October 30, 2018 were used for visual interpretation. Thus, the distribution of thermokarst lakes presents the average areas during the period, which is not affected by the extreme weather events. This has been also explained in the materials and methods section as follows: "To avoid the errors from the period floods and water passage and to obtain the high-quality data with less noise such as clouds, snow and clouds shadow, the Sentinel-2A images on cloudless days during April 1 to October 30, 2018 were used for visual interpretation."

4. Explain in comparison with what data the threshold NDWI = 0.1 was set?

**Response:** The use of the NDWI as 0.1 for mapping the lakes was possible in the Qinghai-Tibetan Plateau (QTP) due to its homogeneous landcover characteristics. Li and Sheng, (2012) has verified the threshold vale of 0.1, which found that the extraction

of the water body is accurate by using the mask of the potential lakes and eliminating the shadow of the mountains. What's more, through the comparison between automatic extraction of remote sensing image and visual interpretation in this study, it is also found that the threshold vale of 0.1 is best to extract the water body.

In the revised version, this has been explained as follows: "The threshold values for a large number of lake sample images were studied, and it was found that the value of 0.1 is enough to extract the area of the potential lake area (Li and Sheng, 2012). By comparing between automatic extraction of remote sensing image and visual interpretation in this study, we confirmed that this value is the best value to extract the water body."

5. How and with what data was visual interpretation carried out to ensure "lake boundary inspection with the highest quality control and consistency" (218)? From which resources and which images of bodies of water were downloaded from online for visual interpretation (220)?

**Response:** Sorry for the confusion. It has been revised as follows: "For the data process, the Sentinel-2A images are firstly processed by color synthesis, then the visual interpretation was carried out according to the information of ground feature spectrum and changes such as the hue and brightness of the images. Three common methods of visual interpretation were used in this study: (1) Direct judgment . It is used to determine the shoreline of the thermokarst lake and identify whether it is collapsing around the shore. (2) Comparison. It is used to determine thermokarst lakes by comparing with the existing materials and field monitoring thermokarst lakes. (3) Logical reasoning. It is used to determine thermokarst lakes through reasoning based on the comprehensive knowledge of geography, hydrology, soil science, and other related subjects. Although this is a time-consuming process, especially for such a large area, this method is useful to recognize the lake boundary with the highest quality control and ensured consistency.".

6. What does three months of visual interpretation mean (223)?

**Response:** The visual interpretation in this study is necessary but a time-consuming and labor-intensive task, and we took three months to complete that. To avoid any confusion, we deleted the three months in the revised version.

7. The data obtained show that even for large TLs, the errors are large (up to 20%) (Table 1), which indicates either the low accuracy of the method or the instability of the TL area, including large ones. For comparison with other results and errors in determining the area of TL according to remote sensing data, the authors are recommended to build a graph of the dependence of the modulus of the average and maximum error (or RMSE) on the area of water bodies for groups, for example, for groups of 400-1000 m, 1000-2000 m ... etc.

**Response:** Thanks for the suggestion. In the present version, the three indicators of $R^2$, MAE and RMSE were added for the verification of extracted data and filed monitoring data. In the results, it has been revised as follows:

We used the coefficient of determination ($R^2$), average absolute error (MAE) and root-mean-square error (RMSE) to present the accuracy assessment. The calculation methods (Draper and Smith, 1998) are as follows Eq. (3-5)

$$R^2 = 1 - \frac{\sum_i (y_i - f_i)^2}{\sum_i (y_i - \hat{y})^2} \tag{3}$$

$$RMSE = \sqrt{\frac{1}{N} \sum_{i=1}^{N} (Z_{oi} - Z_{pi})^2} \tag{4}$$

$$MAE = \frac{1}{N} \sum_{i=1}^{N} |Z_{oi} - Z_{pi}| \tag{5}$$

where $y_i$ is the extracted area of Sentinel-2A thermokarst lakes, $f_i$ is the measured value of UAV, $Z_{oi}$ is the measured value of UAV in the $i$ classification, $Z_{pi}$ is the extracted value of Sentinel-2A, and $N$ is the number of lakes in each classification.

Table 1 Accuracy verification of thermokarst lakes derived from Sentinel-2 data using unmanned aerial vehicle field monitoring

| Area of thermokarst lakes (m²) | N | $R^2$ | RMSE | MAE |
|---|---|---|---|---|
| 400–1000 | 10 | 0.82 | 149.27 | 108.28 |

| 1000–2000 | 12 | 0.83 | 285.00 | 202.61 |
| 2000–5000 | 15 | 0.78 | 599.84 | 493.54 |
| 5000–1000 | 6 | 0.68 | 1765.51 | 1314.39 |
| 10000–20000 | 5 | 0.68 | 1583.53 | 1369.15 |
| 20000–50000 | 3 | 0.99 | 1916.17 | 1769.44 |
| >50000 | 5 | 0.90 | 30929.08 | 22822.81 |

8. The information about the errors (table 1) is in no way connected with the final result – the estimate of the total area of TLs in the QTP area. In this regard, it is not clear why table 1 is needed? At the same time, the information in table 1 can be used to assess the error in determining the total area of water bodies in the region. Any quantitative estimates are always associated with errors.

**Response:** In the present version, the table 1 has been revised (see response to Question 7). We also added the values of $R^2$, MAE and RASE to describe the accuracy of data verification.

9. What information about the number and total area of thermokarst lakes in the region of the entire QTP was previously obtained by other authors, for example, topographic maps of different scales? If there are such data (maps), then it is necessary to show what are the advantages of the data obtained by the authors in comparison with the known data (maps).

**Response:** The previous studies mainly focused on the thermokarst lakes at catchment scale on the middle of QTP, such as the area and changes of thermokarst lakes. The methods were quite different, and the results were obtained from different time. Therefore, it is difficult to compare with our findings. These references (Niu, F., Lin, Z., Hua, L., Lu, J., 2011. Characteristics of thermokarst lakes and their influence on permafrost in Qinghai–Tibet Plateau. Geomorphology 132, 233. Niu, F., Luo, J., Lin, Z., Liu, M., Yin, G., 2014. Morphological Characteristics of Thermokarst Lakes along the Qinghai-Tibet Engineering Corridor. Arctic Antarctic & Alpine Research 46, 963-974.) were cited in the study.

10. In fig. 6 some data do not match the description in the text (268, 269, 270). So in

the text for an altitude of 4750-5000 m, the number of TLs is 59, 314 and the area they occupy is 874.24 km2. On the graph for this height, the number of TLs is more than 70,000, and the area is about 1800 km2.

**Response:** Sorry for the wrong description because there were many statistical charts during our analysis. These chart have been revised as follows:

[Figure]

Figure 6: Number and area of thermokarst lakes at different altitudes on the QTP

11. The map in Fig. 9 does not coincide in contours with the map in Fig. 4.

**Response:** We recreated this figure using the same contours with Fig. 4.

[Figure]

Figure 9: Distribution of thermokarst lakes under different NDVI values

The manuscript can be published after serious revision.

**Response:** We appreciate your constructive comments. We believe the quality of this manuscript has been highly improved. We are happy to address additional concerns.